# Machine Learning for Credit Risk in the Reactive Peru Program: A Comparison of the Lasso and Ridge Regression Models

**Luis Alberto Geraldo-Campos** [1,*] **, Juan J. Soria** [2] **and Tamara Pando-Ezcurra** [1]

1    Dirección de Investigación, Universidad Privada Peruano Alemana, Chorrillos, Lima 15064, Peru; tamara.pando@upal.edu.pe
2    Facultad de Ingeniería de Sistemas y Electrónica, Universidad Tecnológica del Perú, Villa el Salvador, Lima 15842, Peru; c20723@utp.edu.pe
*    Correspondence: luis.geraldo@upal.edu.pe

**Abstract:** COVID-19 has caused an economic crisis in the business world, leaving limitations in the continuity of the payment chain, with companies resorting to credit access. This study aimed to determine the optimal machine learning predictive model for the credit risk of companies under the Reactiva Peru Program because of COVID-19. A multivariate regression analysis was applied with four regressor variables (economic sector, granting entity, amount covered, and department) and one predictor (risk level), with a population of 501,298 companies benefiting from the program, under the CRISP-DM methodology oriented especially for data mining projects, with artificial intelligence techniques under the machine learning Lasso and Ridge regression models, with econometric algebraic mathematical verification to compare and validate the predictive models using SPSS, Jamovi, R Studio, and MATLAB software. The results revealed a better Lasso regression model ($\lambda_{60}$ = 0.00038; RMSE = 0.3573685) that optimally predicted the level of risk compared to the Ridge regression model ($\lambda_{100}$ = 0.00910; RMSE = 0.3573812) and the least squares model with algebraic mathematics, which corroborates that the Lasso regression model is the best predictive model to detect the level of credit risk of the Reactiva Peru Program. The best predictive model for detecting the level of corporate credit risk is the Lasso regression model.

**Keywords:** Lasso model; Ridge model; credits; machine learning; credit risk

## 1. Introduction

The global COVID-19 pandemic has caused a health and economic crisis in individuals and in the various economic and banking sectors (Corredera-Catalán et al. 2021; Hidayat et al. 2021; Ya Liu et al. 2021; Luo 2021; Norden et al. 2021), the latter due to the impact on credit risk on the part of companies (Yin et al. 2022). This is why many countries worldwide have intervened and worked to combat the economic crisis caused by the coronavirus pandemic, especially with small- and medium-sized enterprises (SMEs) being the most vulnerable and important in the business ecosystem (Corredera-Catalán et al. 2021), which is why a great demand for credit has been generated for this sector (Yang et al. 2021) with the support of governments, who have influenced the allocation of credit by banks because the survival of the economy depends on them (Massoc 2021).

Recently, several studies have been carried out on credits and their risk in the face of the global problem caused by COVID-19. In this sense, risk mitigation can be achieved through letters of credit, and financing instruments that provide guarantees for commercial partner services (Crozet et al. 2022). However, the risk of default is latent in enterprises and credit risk is mitigated after the epidemic has been controlled (Yan et al. 2022), while it has been shown that state-owned banks in the periods of the pandemic outbreak managed to reduce their borrowing capacity of SMEs (Liu et al. 2022). It is in this scenario that monetary interventions are associated with lower levels of trade credit, while fiscal interventions

increase due to the use of trade credit (Al-Hadi and Al-Abri 2022). This implies that, COVID-19, produced by SARS CoV-2, significantly affects credit risk and is related to bank capital, total loans, and bank profitability (Riani 2021).

The differential effects of different types of creditor claims on the probability of default and loss of default can show significant intertemporal variation (Heitz and Narayanamoorthy 2021). Meanwhile, companies with higher operational risk tend to adjust trade credit around the target more quickly than those with lower risk exposure (Luo 2021). In the shareholder scenario, the COVID-19 shock was able to increase the credit default swap (CDS) spread, thereby reducing shareholder value for those companies with higher debt rollover risk, however, it is stronger in non-financial, financially constrained, and highly volatile companies (Ya Liu et al. 2021). However, it is the degree of missing data matching, the number of contract defaults, the enforcement rate, the level of business concentration and the amount of administrative penalties that influence SME credit risk, in addition to transactional credit and reputational monitoring (Yang et al. 2021). Therefore, the capacity of governments will depend on the capacity of banks to grant credit to companies (Massoc 2021) and the latter to meet their credit obligations.

Peruvian companies were economically affected due to the social isolation measures established by the executive in the second week of March 2020, in order to face the health care emergency generated by COVID-19, being that companies were subjected to an enormous risk in the continuity of their payment chain (Sampén et al. 2021), so the government established business programs to alleviate the economic havoc wrought by COVID-19 in companies at the national level. One of these programs that had the greatest acceptance and disagreement was the Reactiva Perú Program enacted by Decreto Legislativo No. 1455 (2020) and extended by Decreto Supremo No. 335-2020-EF (2020), which was aimed at companies affected by the COVID-19 health emergency, with the intention of promoting financing to companies facing payments and obligations with their collaborators and suppliers, with the promise of safeguarding the continuity of the payment chain in the country (Decreto Legislativo No. 1455 2020).

So, why should we determine the credit risk in the framework of the Reactiva Peru Program? This question arises due to the existing problem of credit risk for the companies benefiting from the program, given that the financing fund amounted to PEN 60 billion T(8% of GDP), specifically destined to guarantee loans from the Financial System Entities (ESF), administered by the Development Finance Corporation (COFIDE). However, these guarantees may be at risk, since they more reflect the identity of the borrower who determines the value of the loan, but it is the risk of the lender (companies benefited by Reactiva Peru) that may be limited to the borrower's willingness to pay or their inability to meet their obligations, which may be reflected in the short-term in the borrowers (Yan et al. 2022). In this sense, the beneficiary companies are the ones that take the credits and they are the ones that must adequately manage these working capital funds, although their scope of financing is limited to the acquisition of assets, the purchase of shares, bonds, monetary assets, the payment of overdue obligations and not to use it as capital contribution; on the other hand, they have the responsibility and obligation to reactivate the Peruvian economy (COFIDE 2020; Martinez and Pérez 2020).

In the Peruvian case, studies indicate that the companies that have benefited from the Reactiva Perú Program have shown a positive improvement in liquidity to continue with their activities and meet their short-term obligations (Martinez and Pérez 2020; Riani 2021). Likewise, it has been stated that the Reactiva Peru Program has a positive impact on working capital, allowing them to continue with their daily commercial operations of buying and selling (Sudario 2021); in addition, it has been identified in the gray literature that interest rates have been reduced by up to 4.3% and the supply of credit has had an increase of 38% for a certain sector (Quispe 2020), assuming that the program has benefited companies so that they do not go bankrupt and stop generating employment (Monzón et al. 2021). However, the knowledge of and access to the Reactiva Peru Program has been revealed, where it was found that 20% had insufficient knowledge, followed by 75%

with an average level of knowledge and access, 33.8% had access to a loan, and 3.8% benefited from two loans financed by Reactiva Peru, while 36.3% were not able to apply for financing and therefore their economy was affected and disrupted by the impact of COVID-19 (Bocanegra et al. 2021).

A published report stated that 501,298 companies in total (first credit and second credit) were benefited by the Reactiva Peru Program as of 30 October 2020, with a loan amount totaling PEN 57,863,747,358.00 and a covered amount of PEN 52,158,699,017.00 distributed among companies located in 25 departments of Peru (COFIDE 2020; MEF 2020). This is a calculated risk of the Peruvian government, which is currently revealing the effects of the Reactiva Peru Program (Cuadros 2022) and being that many companies are not paying due to the way in which the credits of the Reactiva Peru Program were given, that is, of those companies that were mostly benefited (La República 2022). Therefore, it is worth determining the credit risk through a predictive model of machine learning by means of the Lasso and Ridge multiple regression models, which can be verified with algebraic mathematics by the least squares from the list of beneficiary companies published by the Ministry of Economy and Finance, which allows public decision makers of the credit risk granted to generate strategies and minimize the risks on the part of the beneficiary companies.

## 2. Methodology

This section presents the quantitative analysis of the dataset, using multivariate regression analysis with machine learning techniques under the Lasso and Ridge regression models (Dalgaard 2008; Tan et al. 2019) and verification with algebraic least squares mathematics to compare and validate the models. For this purpose, analysis software such as SPSS, Jamovi, R Studio, and MATLAB were considered, under the CRISP-DM method, especially for data mining projects, which determined the ten-phase approach to determine the best model that predicts the credit risk of the Reactiva Peru Program (Figure 1).

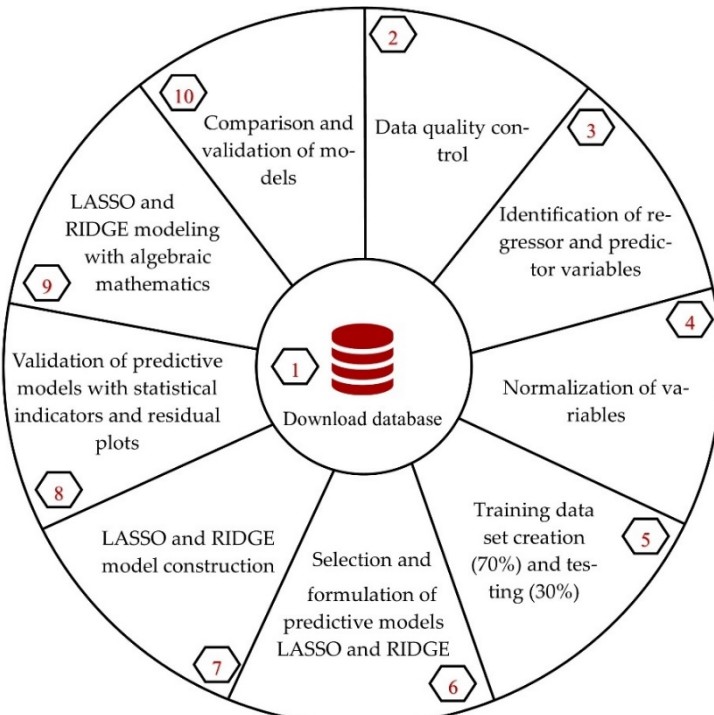

**Figure 1.** The research method.

## 2.1. Download Database

The problem was identified, and the information was verified on the website of the Ministry of Economy and Finance (MEF), where it was possible to access the statistical reports issued and the list of companies benefiting from the Reactiva Peru Program, updated to 30 October 2020. This gave way to downloading the publicly available Excel database. The data can be found at: https://bit.ly/ListadeempresasRP-2020 (accessed on 5 February 2022).

## 2.2. Data Quality Control

A total of 501,298 companies benefited from the Reactiva Peru Program, grouped into microenterprises, small enterprises, medium-sized enterprises, and large enterprises, benefiting 2,561,236 employees (MEF 2020) (See Table 1).

**Table 1.** The distribution of companies that accessed the Reactiva Peru Program according to size and the number of employees.

| Size of Company | No. of Companies | (%) | Amount Placed (Millions of S/) | (%) | No. of Jobs | % |
|---|---|---|---|---|---|---|
| Microenterprise | 445,534 | 88.88% | 8220.7 | 14.21% | 496,191 | 19.37% |
| Small business | 47,234 | 9.42% | 18,477.7 | 31.93% | 667,381 | 26.06% |
| Medium-sized enterprise | 2011 | 0.40% | 2838.2 | 4.90% | 80,329 | 3.14% |
| Large companies | 6519 | 1.30% | 28,327.2 | 48.95% | 1,317,335 | 51.43% |
| Total | 501,298 | 100.00% | 57,863.8 | 100.00% | 2,561,236 | 100.00% |

Source: Adapted from MEF (2020).

After downloading the list of beneficiary companies from the MEF's web portal, a copy of the data was made in Microsoft Excel for efficient quality control. In this process, eight variables were identified: the name of the company, RUC/DNI (Single Register of Taxpayers/National Identity Document), the economic sector, name of the entity granting the loan, name of the second entity granting the loan (companies that received a second loan), loan amount (s/), amount covered (s/), and departments. Of the eight variables identified, four were eliminated: the name of the firm, RUC/DNI, name of the second lending institution, and amount of the loan (s/), because they do not contribute to the main objective of the study. The name of the company and the RUC/DNI were equivalent, and were eliminated because there was no variability (few companies took out two loans), and the name of the second lending institution was eliminated because the study only focused on the level of risk of the companies that took out the first loan granted. The variable "amount of the loan" could have been considered in the present study, but was not taken into account since the amount covered is the most important data for predicting credit risk, so these variables did not contribute to the main objective of the study. It should be noted that prior to the elimination, an attempt was made to analyze these variables so they went through a normalization process, but they lacked this assumption, and since most of them could not be transformed, they were not considered. The following tables show the percentage behavior of the most relevant variables in relation to the beneficiary companies.

Table 2 shows the total number of companies that benefited according to the economic sector. The commerce sector had the highest loan coverage (47.48%), followed by the transportation, storage, and communications sector with 11.90%, equivalent to 59,661 beneficiary companies. Of the 14 economic sectors, the electricity, gas, and water sectors benefited the least from the Reactiva Peru Program, reaching a coverage of 0.14%, equivalent to 717 companies.

**Table 2.** The companies covered by the economic sector.

| Economic Sector * | Covered Companies | % |
|---|---|---|
| Real estate, business, and rental activities (1) | 40,003 | 7.98% |
| Agriculture, livestock, hunting, and forestry (2) | 21,797 | 4.35% |
| Trade (3) | 237,995 | 47.48% |
| Construction (4) | 27,117 | 5.41% |
| Electricity, gas, and water (5) | 717 | 0.14% |
| Teaching (6) | 3094 | 0.62% |
| Hotels and restaurants (7) | 24,567 | 4.90% |
| Manufacturing industry (8) | 48,576 | 9.69% |
| Financial intermediation (9) | 696 | 0.14% |
| Mining (10) | 1350 | 0.27% |
| Fishing (11) | 2942 | 0.59% |
| Social and health services (12) | 5342 | 1.07% |
| Transportation, warehousing and communications (13) | 59,661 | 11.90% |
| Other services (14) | 27,441 | 5.47% |
| Total companies | 501,298 | 100% |

* Numbers in parentheses were assigned by the researchers in alphabetical order.

Figure 2 shows the percentages of the beneficiary companies by department, where companies in Lima benefited the most, covering 30.57% of companies, followed Puno with 7.46%, and the fewest companies covered was in Huancavelica, where only 1728 companies had access, representing 0.34% of companies.

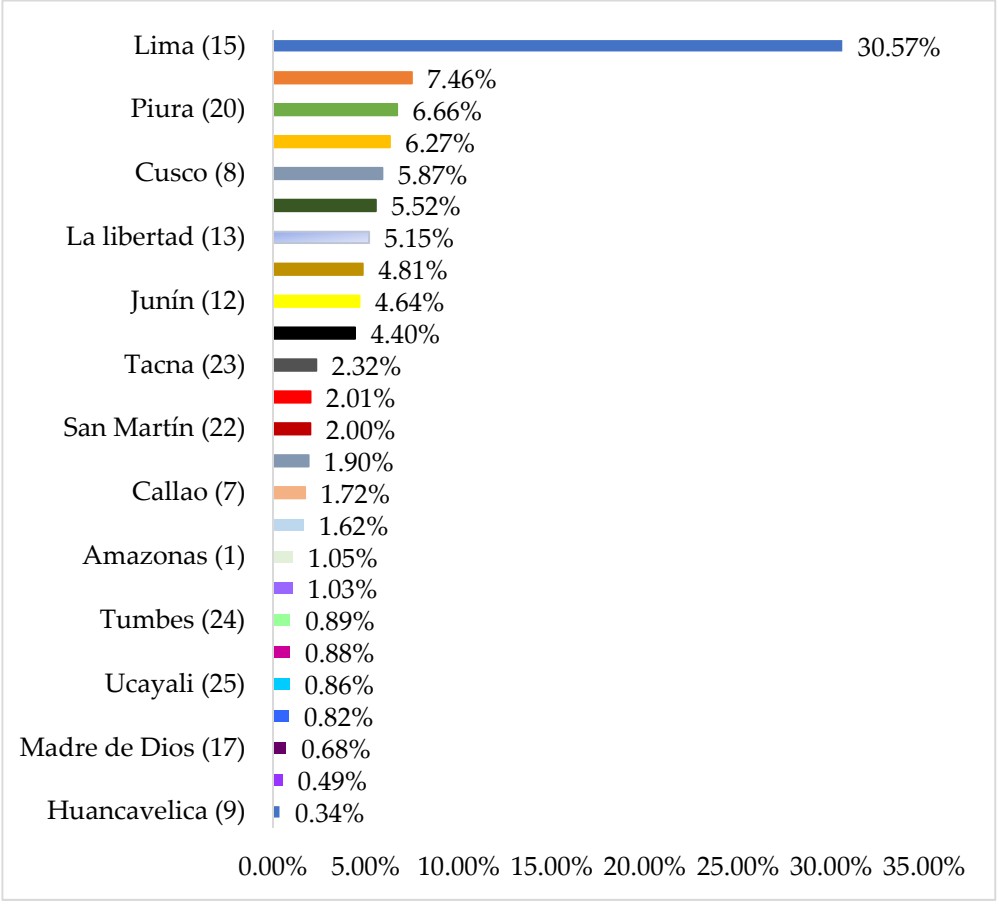

**Figure 2.** The percentage of companies benefited by department. The numbers in parentheses were assigned by the researchers in alphabetical order.

Table 3 shows the percentage results of the companies that accessed credit according to the list of lending institutions. The financial institution that provided the most loans to companies was Mibanco, with a total of 255,671, equivalent to 51.002%, followed by Banco de Crédito BCP with 12.933% of companies that benefited, and the bank with the fewest companies that benefited was Santander Perú S.A., with only nine companies benefiting from the Reactiva Perú Program loan.

**Table 3.** The companies that accessed credit by lending institution.

| Name of the Entity Granting the Loan * | Covered Companies | % |
|---|---|---|
| Crédito BCP (1) | 64,832 | 12.933% |
| Interbank (2) | 19,677 | 3.925% |
| Scotiabank (3) | 12,296 | 2.453% |
| Banco BBVA Perú (4) | 25,101 | 5.007% |
| Comercio (5) | 127 | 0.025% |
| Banco Pichincha (6) | 679 | 0.135% |
| Interamericano (7) | 466 | 0.093% |
| Financiera Crediscotia (8) | 2631 | 0.525% |
| Mibanco (9) | 255,671 | 51.002% |
| Santander Perú S.A. (10) | 9 | 0.002% |
| Financiera Credinka (11) | 4883 | 0.974% |
| Financiera Compartamos (12) | 3073 | 0.613% |
| QAPAQ (13) | 4293 | 0.856% |
| Financiera Efectiva (14) | 29 | 0.006% |
| Financiera Proempresa S.A. (15) | 6849 | 1.366% |
| Financiera Confianza (16) | 269 | 0.054% |
| CMCP Lima (17) | 1999 | 0.399% |
| CMAC Piura (18) | 1514 | 0.302% |
| CMAC Trujillo (19) | 6669 | 1.330% |
| CMAC Arequipa (20) | 22,095 | 4.408% |
| CMAC Sullana (21) | 4424 | 0.883% |
| CMAC Cusco (22) | 19,331 | 3.856% |
| CMAC De Huancayo (23) | 19,101 | 3.810% |
| CMAC De Ica (24) | 2038 | 0.407% |
| CMAC Maynas (25) | 3527 | 0.704% |
| CMAC Tacna (26) | 3697 | 0.737% |
| CRAC Raíz (27) | 15,847 | 3.161% |
| CRAC Prymera (28) | 171 | 0.034% |
| Total companies | 501,298 | 100% |

* The numbers in parentheses were assigned by the researchers in alphabetical order.

### 2.3. Identification of Regressor and Predictor Variables

After quality control of the data and descriptive exploration of the variables, four regressor variables were identified: economic sector, the name of the entity granting the loan, the amount covered (PEN), and department (See Table 4). In this step, a correlative numerical value was assigned to the qualitative variables according to the alphabetical order of their categories, which can be seen in parentheses in Tables 2 and 3, and in Figure 2.

**Table 4.** The symbology of the regressor and predictor variables.

| Symbology | Variable Name | Type of Variable |
|---|---|---|
| X1 | Economic sector | Regressor |
| X2 | Credit granting entity | Regressor |
| X3 | Amount Covered | Regressor |
| X4 | Department | Regressor |
| Y | Level of risk | Predictor |

A logical transformation of the quantity covered was used to generate an ordinal interval variable (considering the levels according to SBS) and create a dummy variable (Pérez 2004; Tsuchiya et al. 2021). The minimum (229.32), maximum (8,500,000), quartile 1 (4890.2), quartile 2 (11,760), and quartile 3 (30,079.7) were considered, which allowed for the categorization of the predictor variable, equivalent to 1 = With potential problems, 2 = Deficient, 3 = Doubtful, and 4 = Lost, according to the levels pre-established by the Superintendency of Banking, Insurance, and AFP (SBS 2019).

An analysis of the level of risk by category in relation to the unstandardized amount covered was carried out by using the multinomial regression technique, resulting in a risk level of 26.12% with potential problems and 25.000% with a loss, followed by 24.951% with a doubtful level, and finally, 23.931% with a deficient level (See Figure 3).

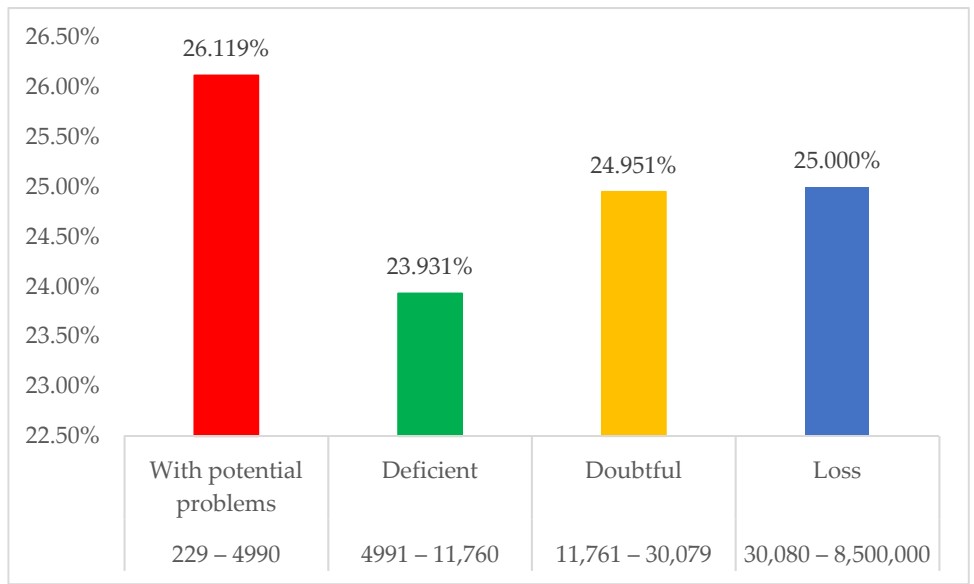

**Figure 3.** The percentage of the level of risk by amount covered.

### 2.4. Normalization of Variables

The present research used data mining with unscaled variables. In this regard, Shanker et al. (1996) suggests that when using data mining and with the application of automatic learning techniques, it is necessary to normalize the characteristics of the variables, since they produce better results in general. In addition, the requirement of the algorithms require the normalization of the data (Atlas et al. 1990), in this case, of the regressors and predictors identified. For this, we proceeded to normalize the data using the min–max normalization technique in order to ensure homogeneity in the variables concentrated in a continuous interval [0; 1] (M-Dawam and Ku-Mahamud 2019) by considering Equation (1):

$$\hat{X}[:,i] = \frac{X[:,i] - \min(X[:,i])}{\max(X[:,i]) - \min(X[:,i])} \tag{1}$$

### 2.5. Training Dataset Creation (70%) and Testing (30%)

The normalization of the variables allowed for the process of creating the training dataset equivalent to 70% (called train) and 30% of the test dataset (called test1) to be used for risk level prediction. These two databases were imported into the R Studio software for their respective analysis, initially verifying the descriptive training data were equal to 350,909 companies, and test1 was equal to 150,389, identifying five study variables in both files.

*2.6. Selection and Formulation of Lasso and Ridge Predictive Models*

2.6.1. Lasso Model Prediction Measures

At the end of the last decade of the last century, the Lasso (Least Absolute Shrinkage and Selection Operator) model was proposed as a method to estimate linear models, with the purpose of minimizing the residual sum of squares, conditional on the sum of the absolute value of the coefficients being less than a constant (Tibshirani 1996). Therefore, small coefficients can be reduced to zero, thus eliminating them from the model, or a small subset can be larger and non-zero (Friedman et al. 2010). This works when the number of variables tends to be large, or in cases when the number of variables is larger than the sample (Hair et al. 2018). Lasso regression and recursive estimations were also performed and the penalty coefficient "$\lambda$" was selected at each recursive step on the basis of cross-validation, focusing on the mean square error (Friedman et al. 2010). However, we defined lambda ($\lambda$) as the weight or regularization parameter assigned to the Lasso and Ridge models (Hastie et al. 2016).

The Lasso regression model represented mathematically (Hastie et al. 2016) has the equation as follows, in addition to the complementary results in Appendix A:

$$\underset{\alpha,\ \beta}{Minimize}\left\{ \frac{1}{2N} \sum_{i=1}^{N}\left( y_i - \alpha - \sum_{j=1}^{p} x_{ij}\beta_j \right)^2 \right\} \tag{2}$$

2.6.2. Ridge Model Prediction Measures

Ridge regression is a particular adaptation of least squares and allows one to address the estimation problem by producing a biased estimator but with small variances (Crocker and Seber 1980). In addition, it allows for data analysis to be performed when multi-collinearity exists and helps to avoid over-fitting (i.e., during the procedure, it removes part of the variance in exchange for a small bias, producing more useful coefficient estimates when such multicollinearity is present) (Frost 2019).

From another point of view, unlike Lasso regression, Ridge regression reduces the coefficients of the correlated predictors, which allows them to borrow the strength of the others. From a Bayesian perspective, the penalty of the Ridge model is appropriate in cases where there are several predictors and they all have non-zero coefficients (i.e., they are drawn from a Gaussian distribution) (Friedman et al. 2010). Furthermore, it is a priority to consider the properties of the Ridge regression mean square error such as the variance and bias of the estimator, the theorem on the mean square function, and the comments made on the mean square error function in the analysis (Crocker and Seber 1980; Hoerl and Kennard 1970). Therefore, the Ridge regression model can be mathematically represented in the following equation:

$$\beta^{ridge} = \underset{\beta\epsilon\mathbb{R}}{argmin}\|Y - X\beta\|_2^2 + \lambda\|\beta\|_2^2 \tag{3}$$

## 3. Results

*3.1. Lasso and Ridge Model Construction*

3.1.1. Lasso Model Results

In Table 5, we can observe the sixty best $\lambda$ contractions for the Lasso model, of which the optimum was $\lambda_{60}$, whose value is equal to 0.00038, being the most efficient of all the lambdas due to its tendency to zero. However, the remaining three ($\lambda_{59}$, $\lambda_{58}$ and $\lambda_{57}$) were analyzed, which contributed less, but allowed for a comparison of the efficiency of the optimal model.

**Table 5.** The top 60 Lasso model lambdas obtained.

| The Top 60 Lasso Model Lambdas | | | | | | | |
|---|---|---|---|---|---|---|---|
| 0.09098 | 0.08290 | 0.07553 | 0.06882 | 0.06271 | 0.05714 | 0.05206 | 0.04744 |
| 0.04322 | 0.03938 | 0.03588 | 0.03270 | 0.02979 | 0.02715 | 0.02473 | 0.02254 |
| 0.02053 | 0.01871 | 0.01705 | 0.01553 | 0.01415 | 0.01290 | 0.01175 | 0.01071 |
| 0.00976 | 0.00889 | 0.00810 | 0.00738 | 0.00672 | 0.00613 | 0.00558 | 0.00509 |
| 0.00463 | 0.00422 | 0.00385 | 0.00351 | 0.00319 | 0.00291 | 0.00265 | 0.00242 |
| 0.00220 | 0.00201 | 0.00183 | 0.00167 | 0.00152 | 0.00138 | 0.00126 | 0.00115 |
| 0.00105 | 0.00095 | 0.00087 | 0.00079 | 0.00072 | 0.00066 | 0.00060 | 0.00055 |
| 0.00050 | 0.00045 | 0.00041 | 0.00038 | | | | |

Table 6 shows the coefficients and the $\lambda$ of the four best Lasso models, where the coefficients of the lambda 60 model stand out, reporting an intercept of 0.51487, a coefficient for the economic sector of 0.05878, a coefficient for the granting entity equal to $-0.19292$, a coefficient for the amount covered equal to 1.29671, and a coefficient for the department equal to 0.03115. In addition, the optimal lambda value equal to 0.00038 and the logarithm of the best lambda equal to $-7.88609$ were corroborated, which is shown as a vertical line in Figure 4 (Lasso_$\lambda_{60}$). Likewise, a root mean squared error (RMSE) equal to 0.3573685 with a coefficient of determination $R^2$ equal to 0.07975 was obtained.

**Table 6.** The validation coefficients of the Lasso regression model.

| Models | Model $\lambda_{60}$ | Model $\lambda_{59}$ | Model $\lambda_{58}$ | Model $\lambda_{57}$ |
|---|---|---|---|---|
| Intercept | 0.51487 | 0.51495 | 0.51504 | 0.51513 |
| Economic sector | 0.05878 | 0.05866 | 0.05853 | 0.05839 |
| Credit granting entity | $-0.19292$ | $-0.19280$ | $-0.19266$ | $-0.19251$ |
| Amount covered | 1.29671 | 1.29624 | 1.29571 | 1.29514 |
| Department | 0.03115 | 0.03101 | 0.03086 | 0.03069 |
| Lambda values | 0.00038 | 0.00041 | 0.00045 | 0.00050 |
| Log/best lambda | $-7.88609$ | $-7.79306$ | $-7.70002$ | $-7.60699$ |
| RMSE | 0.3573685 | | | |
| $R^2$ | 0.07975 | | | |

Figure 4 shows the three best lambda plots of the Lasso model ($\lambda_{59}$, $\lambda_{58}$, and $\lambda_{57}$) plus the cross-validation of the optimal model ($\lambda_{60}$), the latter showing the logarithm of the best lambda as a vertical straight line glued to the left margin (Log.$\lambda_{60}$ = $-7.88609$) obtained from the cross-validation (See Figure 4).

### 3.1.2. Ridge Model Results

Table 7 shows the hundred best contractions $\lambda$ for the Ridge model, of which the optimum is the $\lambda_{100}$ whose value is equal to 0.00910, being the most efficient of all the lambdas because it tends to zero. However, the remaining three ($\lambda_{99}$, $\lambda_{98}$, and $\lambda_{97}$) were analyzed to compare the efficiency of the optimal model.

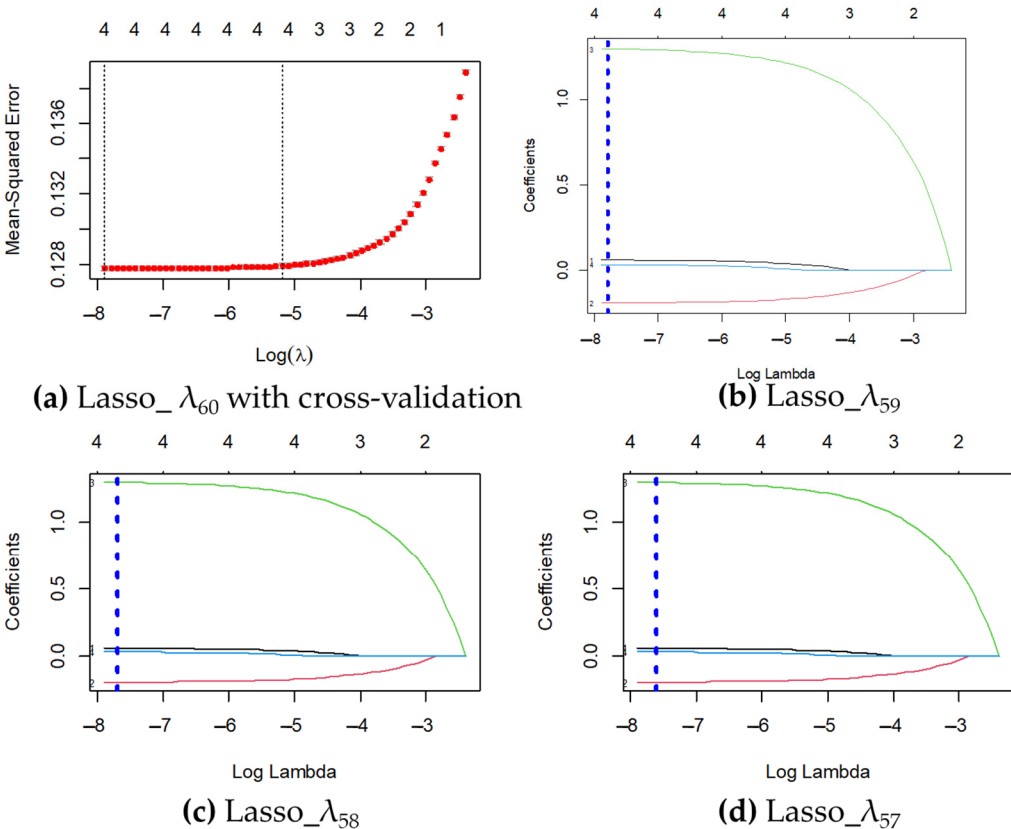

**Figure 4.** The Lasso model and its penalty parameters $\lambda$. (**a**) Mean squared error with cross-validation of the Lasso model based on $\lambda_{60}$ and its $\text{Log}(\lambda_{60})$; (**b**) Coefficients of the Lasso model, with penalties $\lambda_1$ green, $\lambda_2$ red, $\lambda_3$ blue and $\lambda_4$ black based on $\lambda_{59}$ and its $\text{Log}(\lambda_{59})$; (**c**) Coefficients of the Lasso model, with penalty $\lambda_1$ green, $\lambda_2$ red, $\lambda_3$ blue and $\lambda_4$ black based on $\lambda_{58}$ and its $\text{Log}(\lambda_{58})$; (**d**) Coefficients of the Lasso model, with $\lambda_1$ green, $\lambda_2$ red, $\lambda_3$ blue and $\lambda_4$ black based on $\lambda_{57}$ and its $\text{Log}(\lambda_{57})$.

**Table 7.** The top 100 Ridge model lambdas obtained.

| The Top 100 Ridge Model Lambdas | | | | |
|---|---|---|---|---|
| 90.98139 | 82.89885 | 75.53435 | 68.82408 | 62.70994 |
| 57.13896 | 52.06290 | 47.43777 | 43.22353 | 39.38367 |
| 35.88493 | 32.69702 | 29.79230 | 27.14564 | 24.73409 |
| 22.53678 | 20.53468 | 18.71043 | 17.04825 | 15.53373 |
| 14.15376 | 12.89638 | 11.75070 | 10.70680 | 9.75564 |
| 8.88897 | 8.09930 | 7.37978 | 6.72418 | 6.12682 |
| 5.58253 | 5.08660 | 4.63472 | 4.22298 | 3.84782 |
| 3.50599 | 3.19453 | 2.91074 | 2.65216 | 2.41655 |
| 2.20187 | 2.00626 | 1.82803 | 1.66563 | 1.51766 |
| 1.38284 | 1.25999 | 1.14805 | 1.04606 | 0.95314 |
| 0.86846 | 0.79131 | 0.72101 | 0.65696 | 0.59860 |
| 0.54542 | 0.49697 | 0.45282 | 0.41259 | 0.37594 |
| 0.34254 | 0.31211 | 0.28438 | 0.25912 | 0.23610 |
| 0.21512 | 0.19601 | 0.17860 | 0.16273 | 0.14828 |
| 0.13510 | 0.12310 | 0.11217 | 0.10220 | 0.09312 |
| 0.08485 | 0.07731 | 0.07044 | 0.06419 | 0.05848 |
| 0.05329 | 0.04855 | 0.04424 | 0.04031 | 0.03673 |
| 0.03347 | 0.03049 | 0.02778 | 0.02532 | 0.02307 |
| 0.02102 | 0.01915 | 0.01745 | 0.01590 | 0.01449 |
| 0.01320 | 0.01203 | 0.01096 | 0.00999 | 0.00910 |

Table 8 shows the coefficients and the $\lambda$ of the four best Ridge models, where the coefficients of the $\lambda_{100}$ model stand out, which reported an intercept of 0.51408, a coefficient for the economic sector of 0.05849, a coefficient for the granting entity equal to $-0.19071$, a coefficient for the amount covered equal to 1.27321, and a coefficient for the department equal to 0.03198. In addition, the optimal $\lambda$ value was equal to 0.00910 and the logarithm of the best lambda equal to $-4.69969$ was corroborated, which is shown as a vertical line in Figure 4 (Ridge_$\lambda_{100}$). Likewise, a RMSE equal to 0.3573812 was obtained with a coefficient of determination $R^2$ equal to 0.07973.

**Table 8.** The validation coefficients of the Ridge regression model.

| Models | Model $\lambda_{100}$ | Model $\lambda_{99}$ | Model $\lambda_{98}$ | Model $\lambda_{97}$ |
|---|---|---|---|---|
| Intercept | 0.51408 | 0.51408 | 0.51408 | 0.51408 |
| Economic sector | 0.05849 | 0.05835 | 0.05819 | 0.05802 |
| Credit granting entity | $-0.19071$ | $-0.19038$ | $-0.19001$ | $-0.18961$ |
| Amount covered | 1.27321 | 1.27051 | 1.26756 | 1.26433 |
| Department | 0.03198 | 0.03192 | 0.03185 | 0.03178 |
| Lambda values | 0.00910 | 0.00999 | 0.01096 | 0.01203 |
| Log/best lambda | $-4.69969$ | $-4.60665$ | $-4.51362$ | $-4.42058$ |
| RMSE | 0.3573812 | | | |
| $R^2$ | 0.07973 | | | |

Figure 5 shows the three best $\lambda$ plots of the Ridge model ($\lambda_{99}$, $\lambda_{98}$, and $\lambda_{97}$) plus the cross-validation of the optimal model ($\lambda_{100}$), the latter showing the logarithm of the best lambda as a vertical line stuck to the left margin (Log.$\lambda_{100} = -4.69969$) obtained from the cross-validation (See Figure 5).

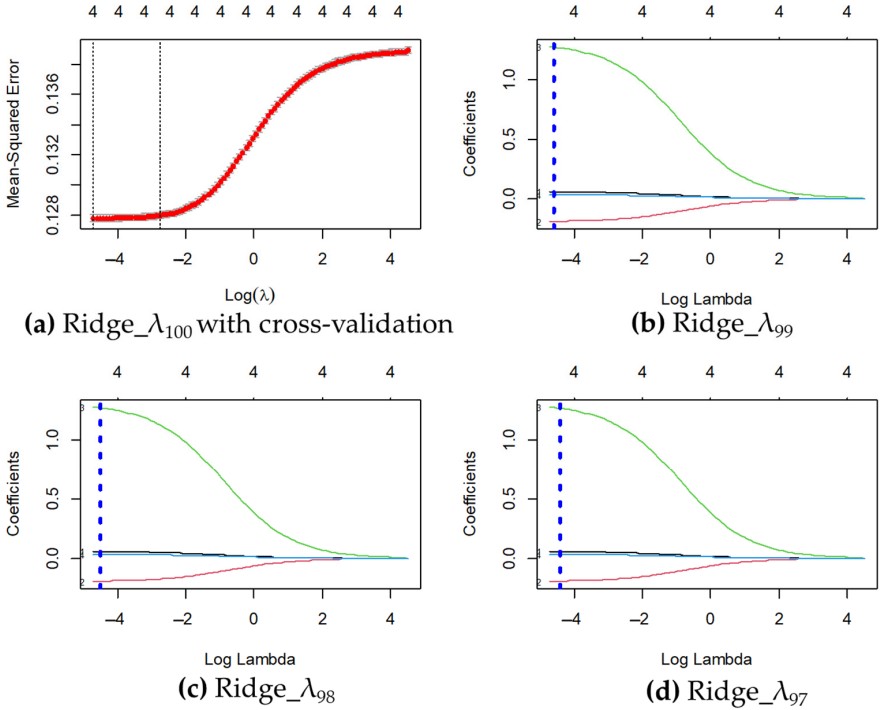

**Figure 5.** Ridge model and its penalty parameters $\lambda$. (**a**) Mean squared error with cross-validation of the Ridge model based on $\lambda_{100}$ and its Log($\lambda_{100}$); (**b**) Ridge model coefficients, with penalty $\lambda_1$ green, $\lambda_2$ red, $\lambda_3$ blue and $\lambda_4$ black based on $\lambda_{99}$ and its Log($\lambda_{99}$); (**c**) Ridge model coefficients, with penalty $\lambda_1$ green, $\lambda_2$ red, $\lambda_3$ blue and $\lambda_4$ black based on $\lambda_{98}$ and its Log($\lambda_{98}$); (**d**) Ridge model coefficients, with penalty $\lambda_1$ green, $\lambda_2$ red, $\lambda_3$ blue and $\lambda_4$ black based on $\lambda_{97}$ and its Log($\lambda_{97}$).

### 3.2. Validation of Predictive Models with Statistical Indicators and Residual Plots

To verify the predictive models, linear regression was used to visualize the most significant p-values of the model. In this sense, two functions were considered in the R Studio software: the first function "lm()" allowed us to perform the linear regression and the second function "summary()" allowed us to visualize the results of the model. To perform the regression with all of the variables, it was formulated with 'Y~.', where the '.' symbol indicates that the rest of the variables in the data were used. This multiple linear regression analysis yielded the summary of results shown in Table 9.

**Table 9.** The ANOVA validation of the model.

|  | Estimate | Std. Error | t Value | Pr(>\|t\|) |
|---|---|---|---|---|
| Intercept | 0.51420 | 0.00264 | 194.897 | $<2 \times 10^{-16}$ *** |
| Economic sector | 0.06411 | 0.00288 | 22.27 | $<2 \times 10^{-16}$ *** |
| Credit granting entity | −0.18914 | 0.00360 | −52.564 | $<2 \times 10^{-16}$ *** |
| Amount covered | 1.33409 | 0.01522 | 87.641 | $<2 \times 10^{-16}$ *** |
| Department | 0.02521 | 0.00356 | 7.093 | $1.32 \times 10^{-12}$ *** |

Note: 0.001 = model significance level. *** = effectiveness of the model being less than 0.001. Residual standard error: 0.3574 on 150,384 degrees of freedom. Multiple R-squared: 0.07981, F-statistic: 3261 on 4 and 150,384. DF, *p*-value: $<2.2 \times 10^{-16}$.

Table 9 shows the approximate estimators obtained from the Lasso and Ridge models as well as the standard deviation, the calculated t student statistic, and the significance of the coefficients of the exposed model, showing, in general, a coefficient of determination $R^2$ equal to 0.07981, which was small but relatively significant.

### 3.3. Lasso and Ridge Modeling with Algebraic Mathematics

3.3.1. Modeling and Verification with Algebraic Mathematics of the Lasso Model

Model Lasso 01 with Lambda 60 is equal to 0.00038:

$$Y_i = \alpha + \beta_1 X_{i1} + \beta_2 X_{i2} + \beta_1 X_{i3} + \beta_2 X_{i4} + \lambda_{60}(|\beta_1| + |\beta_2| + |\beta_3| + |\beta_4|)$$

$$Y_i = 0.51487 + 0.05878 X_{i1} - 0.19292 X_{i2} + 1.29671 X_{i3} + 0.03115 X_{i4} \tag{4}$$

Model Lasso 02 with Lambda 59 is equal to 0.00041:

$$Y_i = \alpha + \beta_1 X_{i1} + \beta_2 X_{i2} + \beta_1 X_{i3} + \beta_2 X_{i4} + \lambda_{59}(|\beta_1| + |\beta_2| + |\beta_3| + |\beta_4|)$$

$$Y_i = 0.51495 + 0.05866 X_{i1} - 0.19280 X_{i2} + 1.29625 X_{i3} + 0.03101 X_{i4} \tag{5}$$

Model Lasso 03 with Lambda 58 is equal to 0.00045:

$$Y_i = \alpha + \beta_1 X_{i1} + \beta_2 X_{i2} + \beta_1 X_{i3} + \beta_2 X_{i4} + \lambda_{58}(|\beta_1| + |\beta_2| + |\beta_3| + |\beta_4|)$$

$$Y_i = 0.51504 + 0.05853 X_{i1} - 0.19266 X_{i2} + 1.29571 X_{i3} + 0.03086 X_{i4} \tag{6}$$

Model Lasso 04 with Lambda 57 is equal to 0.00045:

$$Y_i = \alpha + \beta_1 X_{i1} + \beta_2 X_{i2} + \beta_1 X_{i3} + \beta_2 X_{i4} + \lambda_{57}(|\beta_1| + |\beta_2| + |\beta_3| + |\beta_4|)$$

$$Y_i = 0.51513 + 0.05839 X_{i1} - 0.19251 X_{i2} + 1.29514 X_{i3} + 0.03069 X_{i4} \tag{7}$$

In the research, four Lasso regression models were found with their best penalty coefficients shown in Equations (4)–(7), where the best optimal model obtained a penalty coefficient $\lambda_{60}$ (optimal regularization parameter) that optimized the mean square error. This means that if the economic sector ($X_{i1}$), the lender ($X_{i2}$), the amount covered ($X_{i3}$), and the department ($X_{i4}$) receive a fixed value equal to zero, the average value of the risk level is estimated to be around 51.487%. As the loans are annual loans with historically

low interest rates, between 1% and 2%, the interpretation of the intercept should be taken in moderation. Furthermore, a partial regression coefficient equal to 0.05878 was found, which means that if all other variables are held constant, an increase in the economic sector variable of a company would be found, accompanied by an increase in the average risk level of approximately 5.88, which was equivalent to six beneficiary companies in a given economic sector out of the 14 sectors studied. Similarly, holding all other variables constant, the average risk level decreased by 19.29, which is equivalent to 20 lenders during the lending period. Furthermore, the partial regression coefficient of the amount lent was 1.297, which means that if all of the other variables are held constant, the amount lent by a firm will increase, which will be accompanied by an increase in the average risk level of about PEN 129.671 per loan granted. Finally, holding all other variables constant, the average risk level increased in one department in Peru by 3.115, which means that three departments located in the Peruvian territory benefited significantly from the Reactiva Peru program loan during the loan period.

### 3.3.2. Modeling and Verification of Ridge Model with Algebraic Mathematics

The Ridge regression models obtained with R Studio have the following forms:
Ridge 01 model with Lambda 100 equal to 0.00910:

$$Y_i = 0.51408 + 0.05849X_{i1} - 0.19071X_{i2} + 1.27321X_{i3} + 0.03198X_{i4} \tag{8}$$

Ridge 02 model with Lambda 99 equal to 0.00999:

$$Y_i = 0.51408 + 0.05835X_{i1} - 0.19038X_{i2} + 1.27051X_{i3} + 0.03192X_{i4} \tag{9}$$

Ridge 03 model with Lambda 98 equal to 0.01096:

$$Y_i = 0.51408 + 0.05819X_{i1} - 0.19001X_{i2} + 1.26756X_{i3} + 0.03185X_{i4} \tag{10}$$

Ridge 04 model with Lambda 97 equal to 0.01203:

$$Y_i = 0.51408 + 0.05802X_{i1} - 0.18961X_{i2} + 1.26434X_{i3} + 0.03178X_{i4} \tag{11}$$

Four $\lambda$ of the Ridge regression models were found, with their best penalty coefficients shown in Equations (8)–(11), in which the best optimal model was obtained with a penalty coefficient $\lambda_{100}$ that optimized the mean square error. This means that if the economic sector ($X_{i1}$), granting entity ($X_{i2}$), the amount covered ($X_{i3}$), and the department ($X_{i4}$) collect a fixed value equal to zero, the average value of the risk level is estimated to be around 51.408%. As the loans are annual loans with historically low-interest rates between 1% and 2% of the Reactiva Peru Program, we should consider this mechanical interpretation based on the intercept with caution. This means that holding all other variables constant, we will obtain an increase in the economic sector of a company that will be accompanied by an increase in the average risk level of about 5.85, equivalent to six beneficiary companies. On the other hand, when all of other variables were held constant, the average risk level decreased by 19.071, equivalent to 19 lenders during the lending period. Furthermore, the partial regression coefficient of the amount lent was 1.273 (i.e., if all other variables are held constant, there will be an increase in the amount lent by a firm that will be accompanied by an increase in the average risk level of about PEN 127.321 per loan granted. Finally, if all the other variables were held constant, the average risk level increased in one department in Peru by 3.192, equivalent to three departments located in the Peruvian territory that benefited from the Reactiva Peru program loan during the loan period.

### 3.4. Comparison and Validation of Models

Table 10 shows the comparison of the models' estimators, where the intercept of the multiple regression, Lasso regression, and Ridge estimators were very similar to the algebraic regression estimator. Likewise, for the economic sector variable, the Lasso



and Ridge estimators were quite similar and the multiple regression estimators with the algebraic regression estimators were generally the same. It should be noted that the estimators of the multiple regression of the granting entity differed from the rest of the models, however, its estimators of the amount of credit granted to the companies that accessed the Reactiva Peru Program tended to be very similar in the Lasso, Ridge, and algebraic models, where the multiple regression had a higher estimator than the rest, while the estimators for the department variable varied between 0.02 and 0.3 in the models shown.

**Table 10.** The comparison and validation of models based on the estimators.

| | Multiple Regression Estimation | Lasso Regression Estimation $\lambda_{60}$ | Algebraic Regression with Least Squares Estimation | Ridge Regression Estimation $\lambda_{100}$ |
|---|---|---|---|---|
| Intercept | 0.51420 | 0.51487 | 0.52277 | 0.51408 |
| Economic sector | 0.06411 | 0.05878 | 0.06075 | 0.05849 |
| Credit granting entity | −0.18914 | −0.19292 | −0.19913 | −0.19071 |
| Amount Covered | 1.33409 | 1.29671 | 1.22794 | 1.27321 |
| Department | 0.02521 | 0.03115 | 0.02002 | 0.03198 |

Finally, the comparison and validation of the model estimators allowed us to determine that all of the models presented had good prediction, but the Lasso model was the best, most optimal, and significant at predicting the level of credit risk of the Peruvian companies benefiting from the Reactiva Peru Program due to the best prediction error, an RMSE equal to 0.3573685, and an $R^2$ equal to 0.07975, which was lower than the RMSE of the Ridge model equal to 0.3573812 with an $R^2$ equal to 0.07973.

## 4. Discussion

This study determined the level of credit risk of the Reactiva Peru Program through a Lasso regression model with an optimal penalty coefficient $\lambda_{60}$ equal to 0.00038 with a precision of 0.36. In this regard, Yang et al. (2021), under the application of the Lasso-logistic model with a precision equal to 0.96, showed that the factors that influence the credit risk of small and medium enterprises (SMEs) are the degree of coincidence of missing data, the ratio of contract compliance and the number of defaults of these as well as the degree of business concentration and the number of administrative sanctions. In contrast, Luo (2021) noted that firms with higher operational risk tended to adjust trade credit around the target more quickly than those with lower risk exposure. Therefore, the amount covered by firms benefiting from the Reactiva Peru Program has a higher risk, especially those that obtained a larger amount. In fact, financial institutions should focus on these factors when granting and assessing the level of credit risk in order to make better decisions.

It should be noted that the credit portfolios of banks are often large and complex to visualize; in this sense, Neuberg and Glasserman (2019) mentioned that proper regularization of the portfolio contributes to significantly improve performance, moreover, the application of these methods to credit default swaps allows for margin requirements of the clearing portfolio to be set, and the Lasso method is suitable for estimating the market structure. Liu et al. (2021) noted that the advent of COVID-19 and the shock generated by it have led to an increase in credit default swaps (CDSs), with a significant effect on shareholders, especially in non-financial firms, financially constrained firms, and highly volatile firms. In contrast, in a recent study, Jiang (2022) showed that equity risk has risen to be an important determinant of credit risk.

The comparison and validation of the Lasso regression and Ridge regression models under validation with algebraic mathematics allowed for the validation of the best risk level prediction model, with Lasso being the best prediction model. Contrasting results were found by Wang et al. (2015) when assessing credit risks with the Lasso logistic regression and showed that the proposed algorithm outperformed the most popular credit scoring models such as decision tree, Lasso logistic regression, and random forests. Similarly,

Dai et al. (2021) used several models including using Lasso and recursive feature elimination to predict the bank's credit rating, finding that the SVM model obtained the best accuracy of 86% on the validated dataset and was able to identify that zero and negative revenue days can affect the firm's credit rating. Similarly, Yan et al. (2020) were able to compare machine learning models and found different results to ours, where they mentioned that models incorporating indicator data in multiple time windows conveyed more information in terms of predicting the financial distress compared to existing single-time window models.

In comparison to the aforementioned opposing results, Zhou et al. (2021) agreed with our results, in the sense that they confirmed that the Lasso feature selection method demonstrated a remarkable improvement and outperformed other classifiers. Therefore, they pointed out that the credit score modeling strategy could be used to develop policies, progressive ideas, and operational guidelines for effective credit risk management of loans and other financial institutions. In addition, Ahelegbey et al. (2019) mentioned that the Lasso logistic model for credit scoring led to better identification of the meaningful set of relevant financial characteristics variables, thus producing a more interpretable model, primarily when combined with population segmentation through the factor network approach. However, they emphasized that while the results are promising, they are certainly not definitive.

A similar study on credit risk by Brownlees et al. (2021) proposed a credit risk model, where the interdependence of the default intensity is induced by the exposure to common factors. In contrast, Rao et al. (2020) identified 21 characteristics as a function of rating accuracy, which constitutes the credit risk assessment index system for borrowers in "three rural areas", therefore, considering the owners of these three areas for their rating reduces the volatility of the characteristics and the probability of selection preference and effectively identifies the characteristics that affect the risk rating. In this regard, in the Peruvian case, a large part of the beneficiaries of the Reactiva Peru Program are from rural areas; therefore, it is important to effectively identify the characteristics that may be affecting the non-payment of the loan granted.

However, studies revealed the effectiveness of the Reactiva Peru Program on liquidity to continue its activities and meet their short-term obligations (Martinez and Pérez 2020; Riani 2021). In addition, it has had a positive impact on working capital, allowing them to continue with their daily commercial operations of buying and selling (Sudario 2021) by significantly reducing interest rates by up to 4.3% and increasing the supply of credit by up to 38% in certain sectors (Quispe 2020), with the aim of preventing companies from going bankrupt and ceasing to generate employment (Monzón et al. 2021). However, there are adverse factors such as political, social, and economic factors that may affect the continuity of many companies from benefiting from this program due to the various local and global events that are directly and indirectly related. Therefore, having a model such as the one proposed in this study provides a warning about the level of credit risk, especially for Peruvian companies benefiting from the Reactiva Peru Program.

Based on the above, new lines of research and gaps that arise should be answered with future studies. First, it is necessary to carry out a study with the companies that benefited from a second loan and determine whether they may have a higher level of risk than those that accessed a single loan, under a benchmarking methodology, by considering the data from the first loan in relation to the second block of companies that accessed the second loan. Second, an investigation could be carried out by comparing the credits granted to the Reactiva Peru Program with other programs developed by the Peruvian government such as the Business Support Fund Program for SMEs in the Tourism Sector (FAE-Tourism), the Business Support Program for Micro and Small Enterprises (PAE-Mype), and the National Government Guarantee Program for the Financing of Agricultural Enterprises (FAE-Agro), whose program has had an extension of the credit granting period, under a machine learning methodology with the K-nearest-neighbor (KNN) algorithm, the elastic net model,

or consider the computationally efficient lava prediction model, whose method structure is based on penalization (Chernozhukov et al. 2017).

## 5. Conclusions

The research found that the commerce sector had a loan coverage of 47.48%, followed by the transport, storage, and communications sector with 11.90% and the electricity, gas, and water sector benefited the least with a coverage of 0.14%. In terms of departments, Lima benefited the most with 30.57% of the companies covered, followed by the department of Puno with 7.46%, and the least benefited was Huancavelica with 0.34%. On the other hand, among the financial institutions that granted loans, Mibanco stood out with 51.002% of companies covered, followed by Banco de Crédito BCP with 12.933%, and the bank with the fewest companies was Santander Perú S.A., with only nine companies benefiting from the Reactiva Perú program. The results are conclusive in the sense that companies with lower amounts presented potential problems with a risk level of 26.119%, and companies with amounts higher than 11,761 presented a risk level of 25% and 24.95% with potential and doubtful risks, respectively.

According to the comparison and validation of the model estimators, it can be concluded that all of the models presented in the Results Section have their merits in predicting the level of risk, but the Lasso model was the optimal model for predicting the level of credit risk of the Peruvian companies benefited by the Reactiva Peru Program as its prediction error was better (RMSE = 0.3573685; $R^2$ = 0.07975), being lower than the Ridge model (RMSE = 0.3573812; $R^2$ = 0.07973) with a difference of 0.0000127 and a precision error of 0.00036%. Therefore, the partial regression coefficient of the amount covered was 1.29671437 (i.e., holding all other variables constant will reflect an increase in the amount covered by a company, which is accompanied by an increase in the average risk level of about PEN 129.67144 per loan granted).

Public policies and strategies should be established, considering the Lasso prediction model to control and minimize the risks of the credits granted, in order to minimize the risk of non-payment by the beneficiary companies of the Reactiva Peru Program. The importance of the study lies in presenting the best machine learning predictive model, which is the Lasso model, and encouraging the use and applicability of these models by financial institutions and government agencies to enable them to make better decisions in the economic and business field.

**Author Contributions:** Conceptualization, L.A.G.-C., T.P.-E. and J.J.S.; Methodology, L.A.G.-C. and J.J.S.; Software, L.A.G.-C. and J.J.S.; Validation, L.A.G.-C. and J.J.S.; Formal analysis, L.A.G.-C. and J.J.S.; Investigation, L.A.G.-C. and T.P.-E.; Resources, T.P.-E.; Data curation, L.A.G.-C. and J.J.S.; Writing—original draft preparation, L.A.G.-C., T.P.-E. and J.J.S.; Writing—review and editing, L.A.G.-C.; Visualization, L.A.G.-C. and T.P.-E.; Supervision, L.A.G.-C.; Project administration, L.A.G.-C.; Funding acquisition, T.P.-E. All authors have read and agreed to the published version of the manuscript.

**Funding:** This research was funded by Universidad Privada Peruano Alemana.

**Institutional Review Board Statement:** Not applicable.

**Informed Consent Statement:** Not applicable.

**Data Availability Statement:** Data and other supplementary materials can be found at https://osf.io/qu26s/?view_only=010cc1d0912a4be68ece36b64f33725d (accessed on 7 April 2022).

**Acknowledgments:** We would like to thank the Universidad Privada Peruano Alemana, for the trust placed in us and the facilities provided to carry out this research.

**Conflicts of Interest:** The authors declare no conflict of interest. The funders had no role in the design of the study; in the collection, analyses, or interpretation of data; in the writing of the manuscript, or in the decision to publish the results.

## Appendix A

The Lasso model, considering the variables above-mentioned, is represented by Equation (3):

$$\begin{bmatrix} y_1 \\ y_2 \\ y_3 \\ \vdots \\ y_n \end{bmatrix} = \begin{bmatrix} 1 & x_{11} & x_{12} & x_{13} & x_{14} \\ 1 & x_{21} & x_{22} & x_{23} & x_{24} \\ 1 & x_{31} & x_{32} & x_{33} & x_{34} \\ \vdots & \vdots & \vdots & \vdots & \vdots \\ 1 & x_{k1} & x_{k2} & x_{k3} & x_{k4} \end{bmatrix} \begin{bmatrix} \alpha \\ \beta_1 \\ \beta_2 \\ \beta_3 \\ \beta_4 \end{bmatrix} + \begin{bmatrix} u_1 \\ u_2 \\ u_3 \\ \vdots \\ u_n \end{bmatrix} \tag{A1}$$

$$Y_i = X_{nx5}.\beta_{5x1} + u_i$$

The linear regression model has the form:

$$Y_i = \alpha + \beta_1 X_{i1} + \beta_2 X_{i2} + \beta_3 X_{i3} + \beta_4 X_{i4} + u_i$$

$$u_i = Y_i - \alpha - \beta_1 X_{i1} - \beta_2 X_{i2} - \beta_3 X_{i3} - \beta_4 X_{i4}$$

The least squares technique performs the minimization of the errors, and is represented as:

$$D = \sum_{i=1}^{n} u_i^2$$

In performing the Lasso regression, we added a penalty factor to the least squares, which reduces the loss function $S$ to a minimum value, where:

$$S = \underset{\alpha,\beta_1,\beta_2,\beta_3,\beta_4}{Min} \left[ u_i^2 + \lambda(|\beta_1| + |\beta_2| + |\beta_3| + |\beta_4|) \right]$$

$$S = \underset{\alpha,\beta_1,\beta_2,\beta_3,\beta_4}{Min} \left[ (Y_i - \alpha - \beta_1 X_{i1} - \beta_2 X_{i2} - \beta_3 X_{i3} - \beta_4 X_{i4})^2 + \lambda(|\beta_1| + |\beta_2| + |\beta_3| + |\beta_4|) \right]$$

By applying the least squares optimization conditions, we have:

$$\frac{\partial S}{\partial \alpha} = \frac{1}{2n} \sum_{i=1}^{n} -2(Y_i - \alpha - \beta_1 X_{i1} - \beta_2 X_{i2} - \beta_3 X_{i3} - \beta_4 X_{i4}) = 0$$

where the first normalization equation has the form:

$$\alpha n + \beta_1 \sum_{i=1}^{n} X_{i1} + \beta_2 \sum_{i=1}^{n} X_{i2} + \beta_3 \sum_{i=1}^{n} X_{i3} + \beta_4 \sum_{i=1}^{n} X_{i4} = \sum_{i=1}^{n} y_i$$

The second optimization condition is:

$$\frac{\partial S}{\partial \beta_1} = \frac{1}{2n} \sum_{i=1}^{n} -2x_{i1}(Y_i - \alpha - \beta_1 X_{i1} - \beta_2 X_{i2} - \beta_3 X_{i3} - \beta_4 X_{i4}) + \lambda = 0$$

where the second normalization equation has the form:

$$\alpha \sum_{i=1}^{n} X_{i1} + \beta_1 \sum_{i=1}^{n} X_{i1}^2 + \beta_2 \sum_{i=1}^{n} X_{i1} X_{i2} + \beta_3 \sum_{i=1}^{n} X_{i1} X_{i3} + \beta_4 \sum_{i=1}^{n} X_{i1} X_{i4} = \sum_{i=1}^{n} X_{i1} y_i - \lambda n$$

The third optimality condition has:

$$\frac{\partial S}{\partial \beta_2} = \frac{1}{2n} \sum_{i=1}^{n} -2x_{i2}(Y_i - \alpha - \beta_1 X_{i1} - \beta_2 X_{i2} - \beta_3 X_{i3} - \beta_4 X_{i4}) + \lambda = 0$$

where the third normalization equation has the form:

$$\alpha \sum_{i=1}^{n} X_{i2} + \beta_1 \sum_{i=1}^{n} X_{i2} X_{i1} + \beta_2 \sum_{i=1}^{n} X_{i2}^2 + \beta_3 \sum_{i=1}^{n} X_{i2} X_{i3} + \beta_4 \sum_{i=1}^{n} X_{i2} X_{i4} = \sum_{i=1}^{n} X_{i2} y_i - \lambda n$$

The fourth optimality condition is:

$$\frac{\partial S}{\partial \beta_3} = \frac{1}{2n} \sum_{i=1}^{n} -2x_{i3}(Y_i - \alpha - \beta_1 X_{i1} - \beta_2 X_{i2} - \beta_3 X_{i3} - \beta_4 X_{i4}) + \lambda = 0$$

where the fourth normalization equation has the form:

$$\alpha \sum_{i=1}^{n} X_{i3} + \beta_1 \sum_{i=1}^{n} X_{i3} X_{i1} + \beta_2 \sum_{i=1}^{n} X_{i3} X_{i2} + \beta_3 \sum_{i=1}^{n} X_{i3}^2 + \beta_4 \sum_{i=1}^{n} X_{i3} X_{i4} = \sum_{i=1}^{n} X_{i3} y_i - \lambda n$$

The fifth optimization condition is:

$$\frac{\partial S}{\partial \beta_4} = \frac{1}{2n} \sum_{i=1}^{n} -2x_{i4}(Y_i - \alpha - \beta_1 X_{i1} - \beta_2 X_{i2} - \beta_3 X_{i3} - \beta_4 X_{i4}) + \lambda = 0$$

where the fifth normalization equation has the form:

$$\alpha \sum_{i=1}^{n} X_{i4} + \beta_1 \sum_{i=1}^{n} X_{i4} X_{i1} + \beta_2 \sum_{i=1}^{n} X_{i4} X_{i2} + \beta_3 \sum_{i=1}^{n} X_{i4} X_{i3} + \beta_4 \sum_{i=1}^{n} X_{i4}^2 = \sum_{i=1}^{n} X_{i4} y_i - \lambda n$$

Then, the normal equations of the least squares theory for algebraic multiple regression are:

$$\begin{cases} \alpha n + \beta_1 \sum_{i=1}^{n} X_{i1} + \beta_2 \sum_{i=1}^{n} X_{i2} + \beta_3 \sum_{i=1}^{n} X_{i3} + \beta_4 \sum_{i=1}^{n} X_{i4} = \sum_{i=1}^{n} y_i \\ \alpha \sum_{i=1}^{n} X_{i1} + \beta_1 \sum_{i=1}^{n} X_{i1}^2 + \beta_2 \sum_{i=1}^{n} X_{i1} X_{i2} + \beta_3 \sum_{i=1}^{n} X_{i1} X_{i3} + \beta_4 \sum_{i=1}^{n} X_{i1} X_{i4} = \sum_{i=1}^{n} X_{i1} y_i - \lambda n \\ \alpha \sum_{i=1}^{n} X_{i2} + \beta_1 \sum_{i=1}^{n} X_{i2} X_{i1} + \beta_2 \sum_{i=1}^{n} X_{i2}^2 + \beta_3 \sum_{i=1}^{n} X_{i2} X_{i3} + \beta_4 \sum_{i=1}^{n} X_{i2} X_{i4} = \sum_{i=1}^{n} X_{i2} y_i - \lambda n \\ \alpha \sum_{i=1}^{n} X_{i3} + \beta_1 \sum_{i=1}^{n} X_{i3} X_{i1} + \beta_2 \sum_{i=1}^{n} X_{i3} X_{i2} + \beta_3 \sum_{i=1}^{n} X_{i3}^2 + \beta_4 \sum_{i=1}^{n} X_{i3} X_{i4} = \sum_{i=1}^{n} X_{i3} y_i - \lambda n \\ \alpha \sum_{i=1}^{n} X_{i4} + \beta_1 \sum_{i=1}^{n} X_{i4} X_{i1} + \beta_2 \sum_{i=1}^{n} X_{i4} X_{i2} + \beta_3 \sum_{i=1}^{n} X_{i4} X_{i3} + \beta_4 \sum_{i=1}^{n} X_{i4}^2 = \sum_{i=1}^{n} X_{i4} y_i - \lambda n \\ \qquad\qquad (Ecuaciones\ normales) \end{cases}$$

Arriving in algebraic form, to solve a system of equations of five equations with five variables of the form:

$$\begin{cases} 150389\alpha + 52321.84615\beta_1 + 50901.48148\beta_2 + 1819.60000\beta_3 + 79119.69565\beta_4 = 75479.0000 \\ 52321.84615\alpha + 33634.99408\beta_1 + 18105.6752\beta_2 + 650.91100\beta_3 + 27282.74582\beta_4 = 27135.4121 \\ 50901.48148\alpha + 18105.67521\beta_1 + 27432.10151\beta_2 + 182.93889\beta_3 + 26636.29791\beta_4 = 23004.78435 \\ 1819.60000\alpha + 650.91100\beta_1 + 182.93889\beta_2 + 591.87412\beta_3 + 1016.42942\beta_4 = 1701.47105 \\ 79119.69565\alpha + 27282.74582\beta_1 + 26636.29791\beta_2 + 1016.42942\beta_3 + 51743.12098\beta_4 = 39998.3909 \end{cases}$$

The system in matrix form has the form:

$$\begin{bmatrix} 150389 & 52321.84615 & 50901.48148 & 1819.60000 & 79119.69565 \\ 52321.84615 & 33634.99408 & 18105.6752 & 650.91100 & 27282.74582 \\ 50901.48148 & 18105.67521 & 27432.10151 & 182.93889 & 26636.29791 \\ 1819.60000 & 650.91100 & 182.93889 & 591.87412 & 1016.42942 \\ 79119.69565 & 27282.74582 & 26636.29791 & 1016.42942 & 51743.12098 \end{bmatrix} \begin{bmatrix} \alpha \\ \beta_1 \\ \beta_2 \\ \beta_3 \\ \beta_4 \end{bmatrix} = \begin{bmatrix} 75479.0000 \\ 27135.4121 \\ 23004.78435 \\ 1701.47105 \\ 39998.3909 \end{bmatrix}$$

As the matrix is symmetric and positive definite, the Cholesky technique was applied and the solution for the coefficients of the model was obtained.

$$\begin{cases} \alpha = 0.52277 \\ \beta_1 = 0.06075 \\ \beta_2 = -0.19913 \\ \beta_3 = 1.22794 \\ \beta_4 = 0.02002 \end{cases}$$

The algebraic linear regression model is as follows:

$$Y_{i\ Alg} = 0.52277 + 0.06075X_{i1} - 0.19913X_{i2} + 1.22794X_{i3} + 0.02002X_{i4}$$

The Lasso model with the best Lambda has the form:

$$Y_{i\ Lasso} = 0.51487 + 0.05878X_{i1} - 0.19292X_{i2} + 1.29671X_{i3} + 0.03115X_{i4}$$

Comparing the coefficients of the regression in algebraic form and the Lasso regression, we obtained a difference of 0.0079 equivalent to 0.79% with a precision error of 1.51%; coefficient $\beta_1$ showed a difference of 0.00197 equivalent to 0.20% with a precision error of 0.32%; the coefficient $\beta_2$ showed a difference of 0.00621 equivalent 0.62% with a precision error of 3.12%; coefficient $\beta_3$ showed a difference of 0.066877 equivalent to 6.88% with a precision error of 5.30%; finally, coefficient $\beta_4$ obtained a difference of 0.01113 equivalent to 1.11% with a precision error of 35.73%, which can be corroborated from the augmented Cholesky matrix, where the abscissae, ordinates, and heights were specified (see Figure A1).

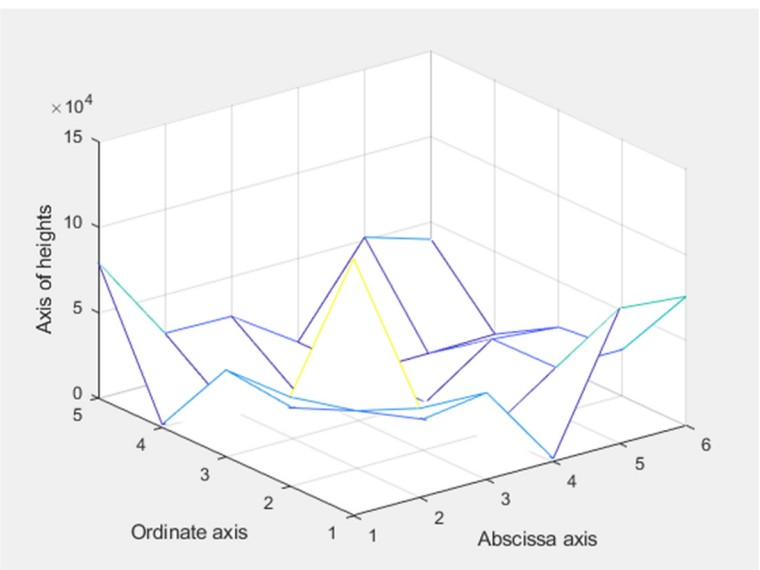

**Figure A1.** The Cholesky augmented matrix chart. This is the matrix graph of the algebraic least squares model, where the lines are shown as points of the matrix surface of the algebraic system, solved by the Cholesky technique.

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
