# Peer review of "Machine Learning for Credit Risk in the Reactive Peru Program: A Comparison of the Lasso and Ridge Regression Models"

_economies, doi:10.3390/economies10080188_

Round 1

Reviewer 1 Report

"Machine Learning predictive model for credit risk of the Reactiva Peru Program as a result of COVID-19 " provides in-depth information about the subject, especially empirical specifications/ calculation.

The manuscript is interesting but some minor improvements are suggested:

1. English language improvements are needed. I suggest authors go through the manuscript carefully and make amendments like:

The following phrases appear with and without a hyphen:

    • ‘least-squares’ / ‘least squares’ 3 times with a hyphen 6 times without
    • ‘que-crea-el-programa’ / ‘que crea el programa’ 1 time with a hyphen 1 time without
    • ‘short term’ / ‘short-term’ 1 time without a hyphen 2 times with
    • ‘trade-credit’ / ‘trade credit’ 1 time with a hyphen 3 times without

These abbreviations appear in two different forms:

    • ‘Lasso’ / ‘LASSO’ Lasso 3 times LASSO 41 times
    • ‘Ridge’ / ‘RIDGE’ Ridge 1 time RIDGE 34 times

2. Abstract: "Determine the optimal Machine Learning predictive model for the credit risk of companies under the Reactiva Peru Program as a result of COVID-19." This statement is not the background. Please amend accordingly. 

3. Line 134: "https://www.youtube.com/watch?v=OgZx_AEl0XM&ab_channel=TuneLyrico" this information could be presented in the form of a footnote.

4. Table 1 needs attention: Please check the spellings in the top row.

5. Line 289: The mathematical results could have been rounded off to particular but identical numbers. The same applies to Table 6. The table is too congested and not easy to go through the results. 

Author Response

Response to Reviewer 1 Comments

Point 1: English language improvements are needed. I suggest authors go through the manuscript carefully and make amendments like: The following phrases appear with and without a hyphen: • ‘least-squares’ / ‘least squares’ 3 times with a hyphen 6 times without • ‘que-crea-el-programa’ / ‘que crea el programa’ 1 time with a hyphen 1 time without • ‘short term’ / ‘short-term’ 1 time without a hyphen 2 times with • ‘trade-credit’ / ‘trade credit’ 1 time with a hyphen 3 times without These abbreviations appear in two different forms: • ‘Lasso’ / ‘LASSO’ Lasso 3 times LASSO 41 times • ‘Ridge’ / ‘RIDGE’ Ridge 1 time RIDGE 34 times

Response 1: • Least-squares was changed to "least squares". • The word 'que-crea-el-programa' with hyphen is from the url of the legislative decree and the one without hyphen 'que crea-el-programa' is from the title of the legislative decree, in this sense neither the title nor the url can be modified, the url because it will not load the page where the legislative decree is hosted. • short-term was changed to "short term". • trade-credit changed to trade credit • LASSO changed to Lasso • RIDGE changed to Ridge

Point 2: Abstract: "Determine the optimal Machine Learning predictive model for the credit risk of companies under the Reactiva Peru Program as a result of COVID-19." This statement is not the background. Please amend accordingly.

Response 2: COVID-19 has caused an economic crisis in the business world, leaving limitations in the continuity of the payment chain, with companies resorting to credit access. This study aimed to determine the credit risk through a predictive model of Machine Learning using multiple regression models, Lasso and Ridge, verifying with algebraic mathematics by least squares the most optimal model for the Reactiva Peru Program.

Point 3: Line 134: "https://www.youtube.com/watch?v=OgZx_AEl0XM&ab_channel=TuneLyrico" this information could be presented in the form of a footnote.

Response 3: Apparently, the URL https://www.youtube.com/watch?v=OgZx_AEl0XM&ab_channel=TuneLyrico is not correct, it opens a video of a singer.

Point 4: Table 1 needs attention: Please check the spellings in the top row.

Response 4: The spelling was improved, the first column did not contain the translation, and the middle rows in the table were removed.

Point 5: Line 289: The mathematical results could have been rounded off to particular but identical numbers. The same applies to Table 6. The table is too congested and not easy to go through the results.

Response 5: All tables, including table 6, were rounded to a maximum of 5 decimal places, except for the RMSE, which allows for the most accurate decision making in predicting the best model for predicting credit risk.

Reviewer 2 Report

Overall, I find the study well done and interesting.  My main comments generally relate to more explanation being needed.  These are the following:

Major comments

  1. Abstract is too long.  It should not be more than 150 words, and it is currently 231 words.

The abstract should generally not include details regarding methods, but rather focus on the research questions and results.

  1. It is unclear to me how on line 311 the intercept α has a precision of 0.79%,, and the coefficients have precisions of XXX. More explanation is needed.
  2. Need to make clear what the meaning of lambda of XX (e.g. 60) means. You have it throughout, but I am never sure what you mean.
  3. Need to explain what an increase in an economic sector of a country refers to. I understand there are 14 sectors.  How does the percent of the sector shown in the Table 2 impact the coefficient?
  4. Line 431 – please explain what “decreases by 19.292089 equivalent to 20 lenders during the lending period.
  5. For those of us not familiar enough, define a penalty coefficient.
  6. Line 542, the prediction error and r-square looks about the same between the LASSO and Ridge model.
  7. Finish the paper with comments regarding the importance of the study and application of these models

Minor comments

  1. Line 45 needs a transition statement
  2. Delete lines 138-144. Too much detail about procedure.
  3. Line 186 what does baremization mean?
  4. Line 319, do you mean multi-colinearity?
  5. Line 538 change, it is concluded to we found that …

Author Response

Response to Reviewer 2 Comments

Point 1: Abstract is too long. It should not be more than 150 words, and it is currently 231 words.

The abstract should generally not include details regarding methods, but rather focus on the research questions and results.

Response 1: The structure of the abstract was considered and reduced to no more than 200 words, as indicated in the template.

Point 2: It is unclear to me how on line 311 the intercept α has a precision of 0.79%,, and the coefficients have precisions of XXX. More explanation is needed.

Response 2: The interpretation of line 311 was improved by specifying the difference and precision error of the coefficients of the algebraic model and Lasso, leaving: Comparing the coefficients of the regression in algebraic form and the Lasso regression we obtained a difference of 0.0079 equivalent to 0.79% with a precision error of 1.51%, the coefficient β_1 a difference of 0.00197 equivalent to 0.20% with a precision error of 0.32%, the coefficient β_2 with a difference of 0.00621 equivalent 0.62% with a precision error of 3. 12%, the coefficient β_3 with a difference of 0.066877 equivalent to 6.88% with a precision error of 5.30%, finally, the coefficient β_4 obtained a difference of 0.01113 equivalent to 1.11% with a precision error of 35.73%, this can be corroborated from the augmented Cholesky matrix, where the abscissae, ordinates, and heights are specified (see fig. 4).

Point 3: Need to make clear what the meaning of lambda of XX (e.g. 60) means. You have it throughout, but I am never sure what you mean.

Response 3: The concept of the penalty lambda coefficient "λ" was clarified, as follows: Lasso regression and recursive estimations are also performed and the penalty coefficient "λ" is selected at each recursive step on the basis of cross-validation, focusing on the mean square error (Friedman et al., 2010). However, we call lambda (λ) the weight or regularization parameter assigned to the Lasso and Ridge model (Hastie et al., 2016).

Point 4: Need to explain what an increase in an economic sector of a country refers to. I understand there are 14 sectors. How does the percent of the sector shown in the Table 2 impact the coefficient?.

Response 4: Some adjustments were made to the wording and interpretation, as follows: In the research, four Lasso regression models were found with their best penalty coefficients shown in equations 5, 6, 7, and 8, where the best optimal model obtained a penalty coefficient λ60 (optimal regularization parameter) that optimized the mean square error. This means that if the economic sector (Xi1), the lender (Xi2), the amount covered (Xi3), and the department (Xi4) receive a fixed value equal to zero, the average value of credit risk is estimated to be around 51.487%. As the loans are annual loans with historically low interest rates, between 1% and 2%, the interpretation of the intercept should be taken with moderation. Furthermore, a partial regression coefficient equal to 0.05878 has been found, which means that if all other variables are held constant, an increase in the economic sector variable of a company is accompanied by an increase in the average credit risk of approximately 5.88, equivalent to 6 beneficiary companies in a given economic sector out of the 14 sectors under study. Similarly, if all other variables are held constant, the average credit risk decreases by 19.29, equivalent to 20 lenders over the lending period. Furthermore, the partial regression coefficient of the amount lent is 1.297, which means that holding all other variables constant will lead to an increase in the amount lent by a firm, which will be accompanied by an increase in the average credit risk of about 129.671 soles per loan granted. Finally, holding all other variables constant, the average credit risk increased in one department in Peru by 3,115 soles, which is equivalent to 3 departments located in the Peruvian territory benefiting from the Reactiva Peru Programme loan during the loan period.

Point 5: Line 431 – please explain what “decreases by 19.292089 equivalent to 20 lenders during the lending period.

Response 5: What decreases is the credit risk in 20 financial institutions, as the values of the other variables remain constant. We indicate that it decreases because the coefficient of the lending institution variable is negative (-0.19251X_i2).

Point 6: For those of us not familiar enough, define a penalty coefficient.

Response 6: This point was answered together with point 3 as follows: Lasso regression and recursive estimations are also performed and the penalty coefficient "λ" is selected at each recursive step on the basis of cross-validation, focusing on the mean square error (Friedman et al., 2010). However, we call lambda (λ) the weight or regularization parameter assigned to the Lasso and Ridge model (Hastie et al., 2016).

Point 7: Line 542, the prediction error and r-square looks about the same between the LASSO and Ridge model.

Response 7: The difference is minimal and has a value of 0.00002 with a precision of 0.025%, which makes the Lasso model have the best R2.

Point 8: Finish the paper with comments regarding the importance of the study and application of these models

Response 8: The wording on the importance and application of the models was improved to read as follows: Public policies and strategies should be established, considering the Lasso prediction model to control and minimize the risks of the credits granted, in order to minimize the risk of non-payment by the beneficiary companies of the Reactiva Peru Programme. The importance of the study lies in presenting the best machine learning predictive model, which is the Lasso model, as well as encouraging the use and applicability of these models by financial institutions and government agencies to enable them to make better decisions in the economic and business field.

Minor comments 1. Line 45 needs a transition statement

Response 1: Line 45 was upgraded to read as follows: The differential effects of different types of creditor claims on the probability of default and loss of default can show significant intertemporal variation.

2. Delete lines 138-144. Too much detail about procedure.

Response 2: These lines describe the phase of the method outlined in Figure 1, therefore, it is necessary to specify the process carried out in the study.

3. Line 186 what does baremization mean?

Response 3: Line 186 was improved, making the paragraph clearer and reading as follows: We then proceeded to create the predictor variable "level of risk" according to the amount covered, for which we calculated the minimum (229.32), maximum (8500000), quartile 1 (4890.2), quartile 2 (11760), and quartile 3 (30079.7). 7), which allowed the categorization of the predictor variable, where 1=With potential problems, 2=Deficient, 3=Doubtful, and 4=Lost according to the levels pre-established by the Su-perintendency of Banks, Insurance and Pension Fund Administrators (SBS) (see Table 5).

4. Line 319, do you mean multi-colinearity?

Response 4: Yes, the error has been corrected

5. Line 538 change, it is concluded to we found that …

Response 5: Line 538 was upgraded, taking into account the reviewer's recommendation, as follows: Public policies and strategies should be established, considering the Lasso prediction model to control and minimize the risks of the credits granted, in order to minimize the risk of non-payment by the beneficiary companies of the Reactiva Peru Programme.

Reviewer 3 Report

The idea is in someways interesting, but I have serious concerns about the way the research is developed and presented.

The main problems are the following

1) The title proposes a predictive model for credit risk. Instead, the paper develops a comparative test on two models, of which the best one scores a r-squared of around 8%, as to say, is able to explicate 8% of the problem.

2) The paper proposes a model for credit risk, but the target variable actually measures  independent variable is reported to be the "amount covered", but no proof or discussion is given on why the "amount covered" is in anyways related to credit risk, which is the risk of a borrower's failure to repay a loan or meet contractual obligations.

3) the "amount covered" variable is recoded into a categorized variable. Why? A quantitative variable contains higher information than a categorized one. This recoding implies a loss of information. Why not keeping the original variable? this needs a strong (by literature or mathematical)  justification.

Less disruptive, but still significant problems refer to the presentation: the excel formula in line 183 makes no sense, as the con tent of cells is not reported; figure 3 cannot be represented by a continuous smoothed line, as it refers to categories; The LASSO model from lines 222 to around 300 seems not developed by the Authors, so better to move it to an appendix; the results include too many decimal values, so the results are not easily readable, which is even worst in table 6; Table 8 is useless; Table 10 is puzzling: if the target variable is the "amount covered", which is the predicted variable tested in this exercise? Figure 6 clearly reports that the residuals are not normally distributed, so the model is misspecified. In section 4 "discussion" no reference is given on how this paper relates to the previous literature, confirming or not the previous results; No significant conclusions are drawn on the title problem, but just on the comparison of LASSO and RIDGE models outcome.

Author Response

Response to Reviewer 3 Comments

Point 1: The title proposes a predictive model for credit risk. Instead, the paper develops a comparative test on two models, of which the best one scores a r-squared of around 8%, as to say, is able to explicate 8% of the problem.

Response 1: A predictive machinne learning model was proposed, but this decision was made on the basis of a comparison of the statistical indicators of the best model predicting credit risk. The coefficient of determination of 8% is minimal, but it allows to explain the problem of the study.

Point 2: The paper proposes a model for credit risk, but the target variable actually measures independent variable is reported to be the "amount covered", but no proof or discussion is given on why the "amount covered" is in anyways related to credit risk, which is the risk of a borrower's failure to repay a loan or meet contractual obligations.

Response 2: In the background search we found few studies that use the Lasso regression model in the analysis of credit risk, so we rely on the comparison with results of similar models but that have used the variables under study that have contributed to decision making. With these comparisons, it has been possible to improve paragraph 1 of the discussion section.

Point 3: the "amount covered" variable is recoded into a categorized variable. Why? A quantitative variable contains higher information than a categorized one. This recoding implies a loss of information. Why not keeping the original variable? this needs a strong (by literature or mathematical) justification.

Response 3: The normalisation of all the variables has been considered because the amount covered had very high values compared to the other variables analysed, in addition, it is based on the machine learning methodology that recommends this technique for a better understanding of statistical processing, which leads to better decision making. However, point 2.4 was improved as follows: With the regressor and predictor variables identified, we proceeded to normalize the data using the Min-Max Normalization technique, in order to have homogeneity in the variables concentrated in a continuous interval [0;1] (M-Dawam & Ku-Mahamud, 2019), considering the following formula:

Point 4: Less disruptive, but still significant problems refer to the presentation: the excel formula in line 183 makes no sense, as the con tent of cells is not reported; figure 3 cannot be represented by a continuous smoothed line, as it refers to categories; The LASSO model from lines 222 to around 300 seems not developed by the Authors, so better to move it to an appendix; the results include too many decimal values, so the results are not easily readable, which is even worst in table 6; Table 8 is useless; Table 10 is puzzling: if the target variable is the "amount covered", which is the predicted variable tested in this exercise? Figure 6 clearly reports that the residuals are not normally distributed, so the model is misspecified. In section 4 "discussion" no reference is given on how this paper relates to the previous literature, confirming or not the previous results; No significant conclusions are drawn on the title problem, but just on the comparison of LASSO and RIDGE models outcome.

Response: Less disruptive, but still significant problems refer to the presentation: the excel formula in line 183 makes no sense, as the con tent of cells is not reported The wording of line 183 was improved to read: We then proceeded to create the predictor variable "level of risk" according to the amount covered, for which we calculated the minimum (229.32), maximum (8500000), quartile 1 (4890.2), quartile 2 (11760), and quartile 3 (30079.7). 7), which allowed the categorization of the predictor variable, where 1=With potential problems, 2=Deficient, 3=Doubtful, and 4=Lost according to the levels pre-established by the Superintendency of Banks, Insurance and Pension Fund Administrators (SBS) (see Table 5). Response: figure 3 cannot be represented by a continuous smoothed line, as it refers to categories Figure 3 shows a smoothed graph with Gaussian trend that the jamovi software shows as a function of the level of risks and the department without normalizing, this result is descriptive and allows to visualize in a general way how the risk is behaving per department based on the amount hedged. Response: The LASSO model from lines 222 to around 300 seems not developed by the Authors, so better to move it to an appendix The Lasso model is described with the formula 2 adapting it to the 5 variables under study describing the least squares method with the objective of using the normal equations that allows to take it to a system of normalised equations and forming a symmetric and positive definite matrix, which, allowed to use the numerical technique of Cholesky to find the coefficients of the Lasso model in algebraic form, that is why it is necessary to register it as an econometric contribution of the study carried out by the authors. Response: the results include too many decimal values, so the results are not easily readable, which is even worst in table 6; Table 8 is useless; Table 10 is puzzling All tables, including table 6, were rounded to a maximum of 5 decimal places, except for the RMSE, which allows for the most accurate decision making in predicting the best model for predicting credit risk. Response: if the target variable is the "amount covered", which is the predicted variable tested in this exercise? The predicted variable is the level of risk. Response: Figure 6 clearly reports that the residuals are not normally distributed, so the model is misspecified The residuals plot shows a low trend of normalization, however, table 10 shows that all variables are significant as their p-values are lower than the significance level α=0.05. It should also be noted that data mining is being analyzed, which justifies flexible normalization for decision making. Response: In section 4 "discussion" no reference is given on how this paper relates to the previous literature, confirming or not the previous results This observation was justified and remedied in the reviser's point 2. Response: No significant conclusions are drawn on the title problem, but just on the comparison of LASSO and RIDGE models outcome. Significant conclusions have been drawn, but they depend on the econometric indicators of the comparison of the models under study. In the title, we indicate a predictive model of machine learning by using the algorithms called Lasso and Ridge regression.

Reviewer 4 Report

Please see below for my comments

Author Response

Response to Reviewer 4 Comments

Referee reports to: Machine learning for credit risk in the Reactiva Peru Programme. 

The author(s) conducted a very thorough analysis, comparing the performance of Lasso and Ridge regression in predicting credit risk. The research methodology is pretty standard and the authors executed it well. I do have some brief comments/suggestions below to improve the readability of the paper.
Point 1: Several clarifications need to be made 
a.    “Predictor” and “Regressor” are used interchangeably in machine learning studies. It is better to call “level of risk”, or Y, as “Outcome Variable”
Response 1.a: Please provide your response for Point 1. (in red)
Changed on the recommendation of the reviewer, standardizing the term "risk level" throughout the document.

b. Table 4 in the current version reads like the author considers only 4 regressor, or predictors. But this cannot be correct. Lasso is a well-known variable selection technique which keeps the most important predictors out of many candidates. If there are only 4 candidates, then there is no point to use Lasso. 
Response 1.b: Please provide your response for Point 1. (in red)
This point was intended to be clarified in section 2.2. Data quality control, because 4 regressor variables were considered, as follows:
After downloading the list of beneficiary companies from the MEF's web portal, a copy of the data was made in Excel for efficient quality control. In this process, eight variables were identified: name of the company, RUC/DNI (Single Register of Taxpayers / National Identity Document), economic sector, name of the entity granting the loan, name of the second entity granting the loan (companies that received a second loan), loan amount (s/), amount covered (s/) and departments. Of the eight variables identified, four were eliminated: name of the firm, RUC/DNI, name of the second lending institution, and amount of the loan (s/), because they do not contribute to the main objective of the study. The name of the company and the RUC/DNI are equivalent, they were eliminated because there was no variability (few companies took out two loans), and the name of the second lending institution was eliminated because the study only focused on the level of risk of those companies that took out the first loan granted, and the variable "amount of the loan" could have been considered in the present study, but was not taken into account since the amount covered is the most important data for predicting credit risk, therefore, these variables do not contribute to the main objective of the study. It should be noted that prior to the elimination, an attempt was made to analyze these variables, so they went through a normalization process, but they lacked this assumption, and since most of them could not be transformed, they were not considered. The following tables show the percentage behavior of the most relevant variables in relation to the beneficiary companies.
c. How is the outcome variable (e.g., level of risk) measured? Is it a discrete credit rating? Or a continuous metric? If the outcome is discrete, then the authors should also use the percentage of correct prediction to compare prediction accuracy other than RMSE.

Response 1.c: Please provide your response for Point 1. (in red)
It is a discrete variable that had to be normalised like the regressor variables, for which the Min-Max Normalization formula was used in point 2.4. However, on the recommendation of reviewer 4, a multinomial risk analysis was performed, considering the level of risk and the amount covered, resulting in its classification as shown in the following figure. The wording and interpretation of this table is shown in the point:
An analysis of the level of risk by category in relation to the unstandardized amount covered was carried out using the multinomial regression technique, resulting in a risk level of 26.12% with potential problems, 25.000% with a loss, followed by 24.951% with a doubtful level, and finally 23.931% with a deficient level.

Point 2: Lasso works well in a “sparse” model in which only a small number of parameters are non-zero. Ridge regression, in contrast, works better when the model is “dense”. That is, most parameters are non-zero but are very small. See Chernozhukov, Hansen and Liao (2017) for a discussion. As such, I conjecture that one reason that Lasso turns out to be a better model is because the risk assessment mechanism under study relies on very few criteria.

Response 2: Please provide your response for Point 2. (in red)
The study by Chernozhukov, Hansen and Liao (2017) was reviewed, where they mention that the Lava model outperforms the sparse and dense estimates provided by the Lasso and Ridge model, considering what the reviewer stated, the use of the data with the Lava prediction model was placed as a recommendation for further study. It was recorded as follows:
Secondly, an investigation can be carried out by comparing the credits granted to the Reactiva Peru Programme with other programmes developed by the Peruvian government, such as the Business Support Fund Programme for SMEs in the Tourism Sector (FAE-Tourism), the Business Support Programme for Micro and Small Enterprises (PAE-Mype) and the National Government Guarantee Programme for the Financing of Agricultural Enterprises (FAE-Agro), whose programmes had an extension of the credit granting period, under a machine learning methodology with the K-Nearest-Neighbor (KNN) algorithm, the Elastic net model or consider the computationally efficient lava prediction model, whose method structure is based on penalization (Chernozhukov et al., 2017).

Point 3: The choice of predictor variables in Table 4 seems a bit ad-hoc. The author should at least justify how did they arrive at the four regressors that they currently use (economic sector, credit granting entity, amount covered and department).
Response 3.
The answer was given in section "b" of point 1.
a.    For example, firm size is an important determinant of credit risk. In Table 1, the authors showed the distribution of company size in the sample. Why not use it as another predictor?
Response 3.a: Please provide your response for Point 3. (in red)
Although we show the size of the company in table 1, it does not appear in the Excel published by the Ministry of Economy and Finance, as it was taken from a published report (that is why the table was cited), so it did not provide an opportunity to take it into account as another predictor, but it seems important to us to include the table, as these are characteristics of the companies that benefited from the Reactiva Peru Programme.
b.    In a recent study, Jiang (2022) shown that equity risk has ascended to be an important determinant for credit risk.
Response 3.b: Please provide your response for Point 3. (in red)
This reference was included in the discussion

Round 2

Reviewer 3 Report

The Authors responses are not satisfactory, all the problems signaled are still there, the revised version is just a superficial polishing of the previous one, no serious revisions have been done.

Author Response

Response to Reviewer 3 Comments

Point 1: The title proposes a predictive model for credit risk. Instead, the paper develops a comparative test on two models, of which the best one scores a r-squared of around 8%, as to say, is able to explicate 8% of the problem.

Response 1:

Considering the reviewer's comment, the title of the manuscript was reworded to read as follows:

Machine learning for credit risk in the Reactiva Peru Programme: a comparison of the Lasso and Ridge regression models.

Point 2: The paper proposes a model for credit risk, but the target variable actually measures independent variable is reported to be the "amount covered", but no proof or discussion is given on why the "amount covered" is in anyways related to credit risk, which is the risk of a borrower's failure to repay a loan or meet contractual obligations.

Response 2:

A response was provided to the reviewer, which included various background information in the discussion section, determining credit risk and in comparison with other existing models.

Point 3: the "amount covered" variable is recoded into a categorized variable. Why? A quantitative variable contains higher information than a categorized one. This recoding implies a loss of information. Why not keeping the original variable? this needs a strong (by literature or mathematical) justification.

Response 3:

It was recoded into a categorized variable as recommended by Perez (2004) and Tsuchiya et al., 2021), as follows:

A logical transformation of the quantity covered was used to generate an ordinal interval variable (considering the levels according to SBS) and create a dummy variable (Pérez, 2004; Tsuchiya et al., 2021). For which the minimum (229.32), maximum (8500000), quartile 1 (4890.2), quartile 2 (11760), and quartile 3 (30079.7) were considered, which allowed the categorization of the predictor variable, equivalent to 1=With potential problems, 2=Deficient, 3=Doubtful and 4=Lost, according to the levels pre-established by the Superintendency of Banking, Insurance and AFP (SBS, 2019).

The rationale for the normalization of all variables is also supported, as follows:

The present research used data mining with unscaled variables. In this regard Shanker et al. (1996) suggests that when using data mining and with the application of automatic learning techniques, it is necessary to normalize the characteristics of the variables, since they produce better results in general, in addition, the requirement of the algorithms require the normalization of the data (Atlas et al., 1990), in this case of the regressors and predictors identified, for this, we proceeded to normalize the data using the Min-Max Normalization technique, in order to have homogeneity in the variables concentrated in a continuous interval [0; 1] (M-Dawam & Ku-Mahamud, 2019), considering equation 1:

Point 4: Less disruptive, but still significant problems refer to the presentation: the excel formula in line 183 makes no sense, as the con tent of cells is not reported; figure 3 cannot be represented by a continuous smoothed line, as it refers to categories; The LASSO model from lines 222 to around 300 seems not developed by the Authors, so better to move it to an appendix; the results include too many decimal values, so the results are not easily readable, which is even worst in table 6; Table 8 is useless; Table 10 is puzzling: if the target variable is the "amount covered", which is the predicted variable tested in this exercise? Figure 6 clearly reports that the residuals are not normally distributed, so the model is misspecified. In section 4 "discussion" no reference is given on how this paper relates to the previous literature, confirming or not the previous results; No significant conclusions are drawn on the title problem, but just on the comparison of LASSO and RIDGE models outcome.

Response: Less disruptive, but still significant problems refer to the presentation: the excel formula in line 183 makes no sense, as the con tent of cells is not reported

The wording of line 183 was improved to read:

A logical transformation of the amount covered was used to generate an ordinal itervalue variable (considering the levels according to SBS) and create a dummy variable (Pérez, 2004). For which the minimum (229.32), maximum (8500000), quartile 1 (4890.2), quartile 2 (11760) and quartile 3 (30079.7) were considered, which allowed the categorization of the predictor variable, equivalent to 1=With potential problems, 2=Deficient, 3=Doubtful and 4=Lost according to the levels pre-established by the Superintendency of Banks, Insurance and Pension Fund Administrators (SBS, 2019).

Response: figure 3 cannot be represented by a continuous smoothed line, as it refers to categories

At the recommendation of the reviewer, Figure 3 and its interpretation were deleted.

Response: The LASSO model from lines 222 to around 300 seems not developed by the Authors, so better to move it to an appendix

At the reviewer's suggestion, the mathematical structure and figure 5 were moved to an Appendix, taking into account that it is mathematics developed by the authors.

Response: the results include too many decimal values, so the results are not easily readable, which is even worst in table 6; Table 8 is useless; Table 10 is puzzling

All tables were rounded to a maximum of 5 decimal places, with the exception of the RMSE, which is what allows the most accurate decision to be made in predicting the best model for predicting credit risk.

Response: if the target variable is the "amount covered", which is the predicted variable tested in this exercise?

The predicted variable is the level of risk.

Response: Figure 6 clearly reports that the residuals are not normally distributed, so the model is misspecified

The residuals plot shows a low normalisation trend, however, table 9 shows that all variables are significant as their p-values are lower than the significance level α=0.05 despite the low normality of the variables. However, in order not to enter into controversy we proceeded to remove the figure of residuals, taking into account that we are analysing a data mining that justifies that the normalisation is flexible for decision making.

Response: In section 4 "discussion" no reference is given on how this paper relates to the previous literature, confirming or not the previous results

Comparisons were made with previous results, leaving the discussion section as follows:

This study determined the credit risk of the Reactiva Peru Programme through a Lasso regression model with an optimal penalty coefficient λ60 equal to 0.00038 with a precision of 0.36. In this regard, Yang et al. (2021) under the application of the Lasso-logistic model with a precision equal to 0.96, evidenced that the factors that influence the credit risk of Small and Medium Enterprises (SMEs) are: the degree of coincidence of missing data, the ratio of contract compliance and the number of defaults of these, also include the degree of business concentration and the number of administrative sanctions. In contrast, Luo (2021) notes that firms with higher operational risk tend to adjust trade credit around the target more quickly than those with lower risk exposure. Therefore, the amount covered by firms benefiting from the Reactiva Peru programme has a higher risk, especially those that obtained a larger amount. In fact, financial institutions should focus on these factors when granting and assessing credit risk for better decision making.

It should be noted that banks' credit portfolios are usually large and complex to visualize, in that sense, Neuberg & Glasserman (2019) mention that an adequate regularisation of the portfolio contributes to significantly improve performance, likewise, the application of these methods to credit default swaps allows establishing margin requirements of the clearing portfolio, and the Lasso method is suitable for estimating the market structure. Liu et al. (2021) note that the advent of COVID-19 and the shock generated by it have led to an increase in credit default swaps (CDS), with a significant effect on shareholders, especially in non-financial firms, financially constrained firms, and highly volatile firms.

The comparison and validation of the Lasso regression and Ridge regression models under validation with algebraic mathematics allowed validation of the best predictive model for credit risk, with Lasso being the best predictor model. Opposite results were found by Wang et al. (2015) when assessing credit risks with Lasso logistic regression and showed that the proposed algorithm outperforms the most popular credit scoring models, such as decision tree, Lasso-logistic regression, and random forests. Similarly, Dai et al. (2021) use several models including using Lasso and recursive feature elimination to predict the bank's credit rating, finding that the SVM model obtains the best accuracy of 86% on the validated dataset and was able to identify that zero and negative revenue days can affect the firm's credit rating. Similarly, Yan et al. (2020) were able to compare machine learning models, finding different results to ours, where they mention that models incorporating indicator data in multiple time windows convey more information in terms of predicting financial distress compared to existing single time window models.

In comparison to the aforementioned opposing results, Zhou et al. (2021) agree with our results, in the sense that they confirm that the Lasso feature selection method, demonstrates remarkable improvement and outperforms other classifiers, therefore, they point out that the credit score modeling strategy can be used to develop policies, progressive ideas, operational guidelines for effective credit risk management of loans and other financial institutions. Add to this Ahelegbey et al. (2019) mentioning that the Lasso logistic model for credit scoring leads to better identification of the meaningful set of relevant financial characteristics variables, thus producing a more interpretable model, primarily when combined with population segmentation through the factor network approach. However, they emphasize that the results are promising, but certainly not definitive.

A similar study on credit risk by Brownlees et al. (2021) proposed a credit risk model, where the interdependence of default intensity is induced by exposure to common factors. In contrast, Rao et al. (2020) identified 21 characteristics as a function of rating accuracy, which constitutes the credit risk assessment index system for borrowers in "three rural areas", therefore, considering the owners of these three areas for their rating reduces the volatility of the characteristics and the probability of selection preference and effectively identifies the characteristics that affect the risk rating. In this regard, in the Peruvian case, a large part of the beneficiaries of the Reactiva Peru Programme are from rural areas; therefore, it is important to effectively identify these characteristics that may be affecting the non-payment of the loan granted.

However, studies revealed the effectiveness of the Reactiva Peru Programme on liquidity to continue its activities and meet short-term obligations (Martinez & Pérez, 2020; Riani, 2021). In addition, it has had a positive impact on working capital, allowing them to continue with their daily commercial operations of buying and selling (Sudario, 2021), significantly reducing interest rates by up to 4.3% and an increase in the supply of credit by up to 38% in certain sectors (Quispe, 2020), with the aim of preventing companies from going bankrupt and ceasing to generate employment (Monzón et al., 2021). However, there are adverse factors such as political, social, and economic factors that may affect the continuity of many companies benefiting from this programme, due to the various local and global events that are directly and indirectly related. Therefore, having a model such as the one proposed in this study makes it possible to alert credit risk, especially for Peruvian companies benefiting from the Reactiva Peru programme.

Based on the above, new lines of research and gaps arise that should be filled with future research. In the first place, there is a need to carry out a study with the companies that benefited from a second loan and determine whether they may have much more credit risk than those that accessed a single loan, under a comparative study methodology, considering the data from the first loan in relation to the second block of companies that accessed the second loan. Secondly, research can be carried out by comparing the loans granted to the Reactiva Perú Program with other programs developed by the Peruvian government, such as the Business Support Fund Program for Mypes in the Tourism Sector (FAE-Turismo), the Business Support Program for Micro and Small Enterprises (PAE-Mype) and the National Government Guarantee Program for Agricultural Business Financing (FAE-Agro), whose programs had an extension of the term for granting credits, under a machine learning methodology with the K-Nearest-Neighbor (KNN) algorithm and Elastic net model.

Response: No significant conclusions are drawn on the title problem, but just on the comparison of LASSO and RIDGE models outcome.

The title was changed based on the conclusions at the suggestion of the reviewer.